# Biologically plausible learning in recurrent neural networks reproduces neural dynamics observed during cognitive tasks

**Thomas Miconi***

The Neurosciences Institute, California, United States

**Abstract** Neural activity during cognitive tasks exhibits complex dynamics that flexibly encode task-relevant variables. Chaotic recurrent networks, which spontaneously generate rich dynamics, have been proposed as a model of cortical computation during cognitive tasks. However, existing methods for training these networks are either biologically implausible, and/or require a continuous, real-time error signal to guide learning. Here we show that a biologically plausible learning rule can train such recurrent networks, guided solely by delayed, phasic rewards at the end of each trial. Networks endowed with this learning rule can successfully learn nontrivial tasks requiring flexible (context-dependent) associations, memory maintenance, nonlinear mixed selectivities, and coordination among multiple outputs. The resulting networks replicate complex dynamics previously observed in animal cortex, such as dynamic encoding of task features and selective integration of sensory inputs. We conclude that recurrent neural networks offer a plausible model of cortical dynamics during both learning and performance of flexible behavior.

*For correspondence: thomas. miconi@gmail.com

**Competing interests:** The author declares that no competing interests exist.

## Introduction

Recent evidence suggests that neural representations are highly dynamic, encoding multiple aspects of tasks, stimuli, and commands in the joint fluctuating activity of interconnected populations of neurons, rather than in the stable activation of specific neurons (*Meyers et al., 2008*; *Barak et al., 2010*; *Stokes et al., 2013*; *Churchland et al., 2012*; *Raposo et al., 2014*). Models based on recurrent neural networks (RNN), operating in the near-chaotic regime, seem well-suited to capture similar dynamics (*Jaeger, 2001*; *Maass et al., 2002*; *Buonomano and Maass, 2009*; *Sussillo and Abbott, 2009*). For this reason, such models have been used to investigate the mechanisms by which neural populations solve various computational problems, including working memory (*Barak et al., 2013*; *Rajan et al., 2016*), motor control (*Sussillo et al., 2015*; *Laje and Buonomano, 2013*; *Hennequin et al., 2014*), and perceptual decision-making (*Mante et al., 2013*).

However, the methods commonly used to train these recurrent models are generally not biologically plausible. The most common training methods are based on supervised learning, in which a non-biological algorithm (usually a form of backpropagation or regression) minimizes the difference between the network's output and a target output signal (*Pearlmutter, 1995*; *Jaeger, 2001*; *Sussillo and Abbott, 2009*; *Song et al., 2016*; *Rajan et al., 2016*). Besides the non-biological nature of these algorithms, the requirement for a constant supervisory signal is in stark contrast with most behavioral tasks, in which the only source of information about performance are temporally sparse rewards that are usually delayed with regard to the actions that caused them.

A more biologically plausible form of learning is reward-modulated Hebbian learning: during ongoing activity, each synapse accumulates a *potential* weight change according to classical Hebbian learning, by multiplying pre- and post-synaptic activities at any time and accumulating this product over time. These potential weight changes are then multiplied by a global reward signal,

which determines the *actual* weight changes. This method, inspired by the effect of dopamine on synaptic plasticity, has been successfully demonstrated and analyzed in feedforward or weakly connected spiking (*Izhikevich, 2007*; *Florian, 2007*; *Frémaux et al., 2010*) and firing-rate (*Soltoggio and Steil, 2013*) networks. However, simple reward-modulated Hebbian learning does not work for strongly-connected recurrent networks that can generate complex trajectories of the type discussed here (*Fiete et al., 2007*).

A method that successfully trains arbitrary recurrent networks is the so-called *node-perturbation* method (*Fiete et al., 2006*, *2007*). This method consists in applying small perturbations to neural activity, then calculating potential weight changes by multiplying the 'normal' (non-perturbative) inputs by the perturbations (rather than by post-synaptic output, as in Hebbian learning). These potential weight changes are then multiplied by a reward signal to provide the final weight changes. This method was successfully applied to feedforward networks to model birdsong learning (*Fiete et al., 2007*) and our own previous results show that it is also successful when applied to chaotic recurrent neural networks (*Miconi, 2014*). Interestingly, this method is largely similar to the well-known REINFORCE algorithm, which is widely used in reinforcement learning (*Williams, 1992*; *Mnih et al., 2014*; *Peters and Schaal, 2008*; *Kober et al., 2013*) (see in particular Eq. 11 in *Williams, 1992*).

However, node perturbation is non-Hebbian (since it multiplies two types of inputs, rather than pre- and post-synaptic activities) and requires information that is not local to the synapse (namely, the perturbatory inputs, which must somehow be kept separate from the 'normal' inputs). Thus, it is not obvious how node-perturbation could be implemented in biological neural networks. Legenstein and colleagues (*Hoerzer et al., 2014*; *Legenstein et al., 2010*) showed that, under certain conditions, node-perturbation could be made more biologically plausible by leveraging moment-to-moment fluctuations in post-synaptic activity: by keeping a running average of recent activity and subtracting it from the current instantaneous response at any time, we obtain a 'high-pass' filtered trace of post-synaptic activity, which can be used as a proxy for the exploratory perturbations of post-synaptic activity. This can then be multiplied by the pre-synaptic inputs, and the final accumulated product is then modulated by a reward signal to recreate the node-perturbation method in a more biologically plausible, Hebbian manner (i.e. as a product of pre-synaptic and post-synaptic activities rather than between two input sources) (*Legenstein et al., 2010*). This method can successfully train chaotic recurrent neural networks (*Hoerzer et al., 2014*). Unfortunately, this method critically requires an instantaneous, real-time continuous reward signal to be provided at each point in time. The continuous, real-time reward signal is necessary to allow the subtraction method to extract task-relevant information, and to counter the effect of spurious deviations introduced by the running-average subtraction process (see Analysis). This is in contrast with most tasks (whether in nature or in the laboratory), which only provide sparse, delayed rewards to guide the learning process.

In summary, to our knowledge, there is currently no biologically plausible learning algorithm that can successfully train chaotic recurrent neural networks with realistic reward regimes.

Here we introduce a novel reward-modulated Hebbian learning rule that can be used to train recurrent networks for flexible behaviors, with reward occurring in a delayed, one-time fashion after each trial, as in most animal training paradigms. This method is Hebbian and uses only synapse-local information, without requiring instantaneous reward signals (see Materials and methods and Analysis). We apply our method to several tasks that require flexible (context-dependent) decisions, memory maintenance, and coordination among multiple outputs. By investigating the network's representation of task-relevant aspects over time, we find that trained networks exhibit complex dynamics previously observed in recordings of animal frontal cortices, such as dynamic encoding of task features (*Meyers et al., 2008*; *Stokes et al., 2013*; *Jun et al., 2010*), switching from stimulus-specific to response-specific representations (*Stokes et al., 2013*), and selective integration of sensory input streams (*Mante et al., 2013*). We conclude that recurrent networks endowed with reward-modulated Hebbian learning offer a plausible model of cortical computation and learning, capable of building networks that dynamically represent and analyze stimuli and produce flexible responses in a way that is compatible with observed evidence in behaving animals.

## Results

### Description of the learning rule

Here we provide an overview of the networks and plasticity rule used in this paper. A full description is provided in Materials and methods. Furthermore, we provide an extensive discussion of the mechanisms underlying the rule in the Analysis section. Note that all the software used in this paper is available at http://github.com/ThomasMiconi/BiologicallyPlausibleLearningRNN.

Our model networks are fully-connected continuous-time recurrent neural networks operating in the early chaotic regime, which allows them to autonomously generate rich dynamics while still being amenable to learning (*Sompolinsky et al., 1988*; *Sussillo and Abbott, 2009*; *Jaeger, 2001*; *Maass et al., 2002*). In most simulations, we use a canonical model in which responses are signed, and each neuron can send both excitatory and inhibitory connections (see Materials and methods). However, in the last section of Results, we build on recent work (*Mastrogiuseppe and Ostojic, 2016*) to test the rule on a recurrent network with nonnegative activations and separate populations of strictly excitatory and strictly inhibitory neurons, which still generates ongoing chaotic activity.

In all simulations, one or more neurons in the network are arbitrarily designated as the 'output' neurons, and their responses at any given time are used as the network's response (these neurons are otherwise identical to all others). For the simulations reported here, networks include 200 neurons (400 for the motor control task).

We now briefly describe the learning rule that trains the network's connectivity (a more complete description is provided in Materials and methods). First, in order to produce exploratory variation in network responses across trials, each neuron $i$ in the network occasionally receives a random perturbation $\Delta_i(t)$ to its current excitation (perturbations are applied in all simulations, both in learning and in testing/decoding). During a trial, at every time step, every synapse from neuron $j$ to neuron $i$ accumulates a *potential* Hebbian weight change (also called eligibility trace [*Izhikevich, 2007*]) according to the following equation:

$$e_{i,j}(t) = e_{i,j}(t-1) + S\big(r_j(t-1) * (x_i(t) - \bar{x}_i)\big) \tag{1}$$

where $r_j$ represents the output of neuron $j$, and thus the current input at this synapse. $x_i$ represents the current excitation (or potential) of neuron $i$ (see Materials and methods) and $\bar{x}_i$ represents a short-term running average of $x_i$, and thus $x(t) - \bar{x}$ tracks the fast fluctuations of neuron output. Thus, this rule is essentially Hebbian, based on the product of inputs and output fluctuations. Importantly, $S$ is a monotonic, *supralinear* function; in this paper, we simply used the cubic function $S(x)=x^3$, though the particular choice of function is not crucial as long as it is supralinear (see Materials and methods and Analysis). Note that the eligibility trace for any synapse is accumulated over the course of a trial, with each new timestep adding a small increment to the synapse's eligibility trace (potential weight change).

At the end of each trial, a certain reward R is issued to the network, based on the network's performance for this trial as determined by the specific task. From this reward, the system computes a *reward prediction error* signal, as observed in physiological experiments, by subtracting the expected reward for this trial $\bar{R}$ (generally a running average of previously received rewards for the same type of trial; see Materials and methods) from the actually received reward R. This reward-prediction signal is used to modulate the eligibility trace, producing the actual weight change:

$$\Delta J_{i,j} = \eta e_{i,j}(R - \bar{R}) \tag{2}$$

where η is a learning rate constant. Together, *equations 1 and 2* fully determine the learning rule described here. See Materials and methods for a complete description.

### Task 1: Delayed nonmatch-to-sample task

The first task considered here is a simple delayed nonmatch-to-sample problem (*Figure 1*). In every trial, we present two brief successive inputs to the network, with an intervening delay. Each input can take either of two values, labelled A and B respectively. The task is to determine whether the two successive inputs are identical (AA or BB), in which case the network should output −1; or

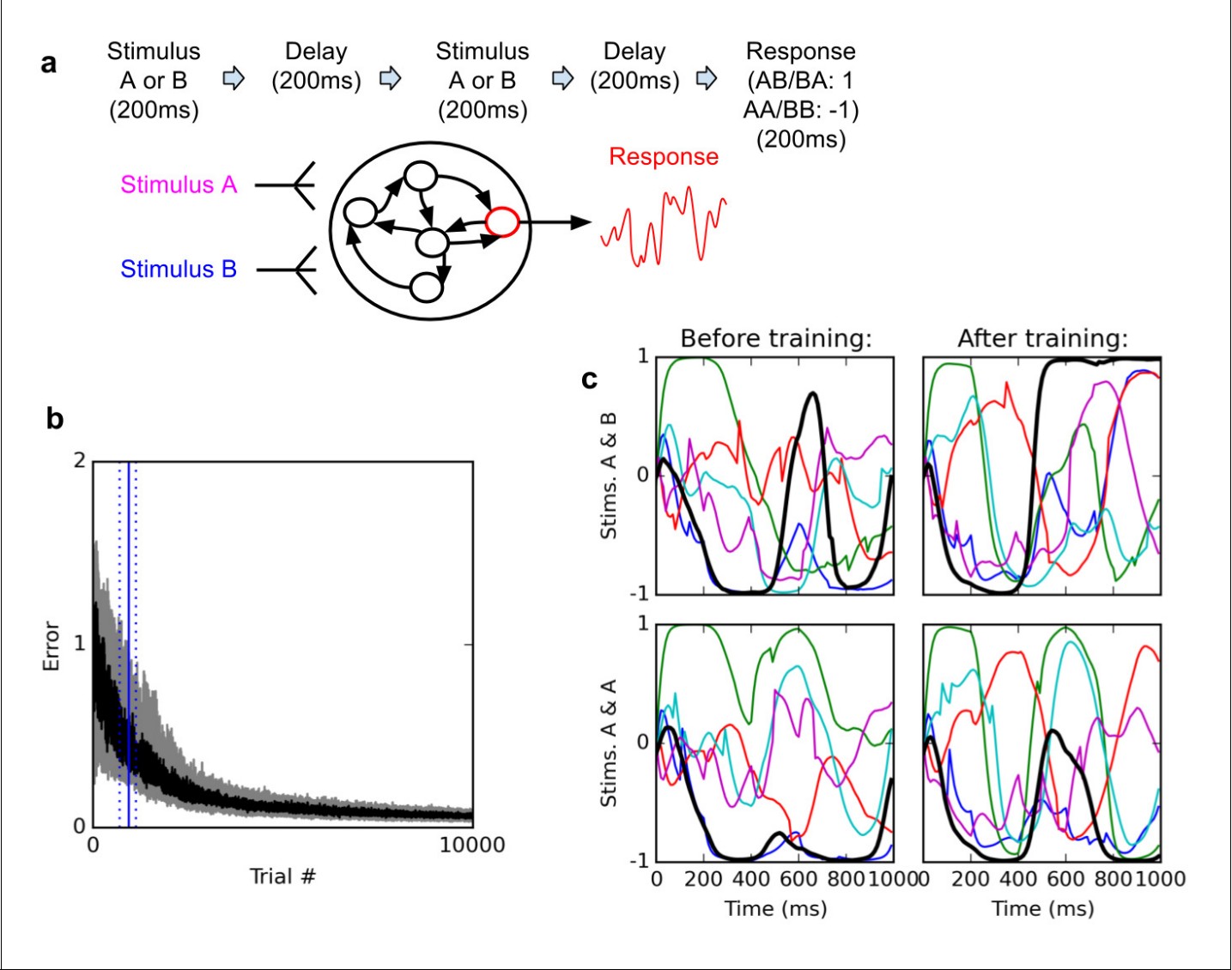

**Figure 1.** Delayed nonmatch-to-sample task. (A) (top): task description. The network is exposed to two successive stimuli, with an intervening delay. The task is to produce output −1 if the two stimuli were identical (AA or BB), or 1 if they were different (AB or BA); the output of the network is simply the activity of one arbitrarily chosen 'output' neuron, averaged over the last 200 ms of the trial. (B) (bottom left): time course of trial error (mean absolute difference between output neuron response and correct response over the last 200 ms of each trial) during learning over 10000 trials (dark curve: median over 20 runs; gray area: inter-quartile range). The solid vertical line indicates the median number of trials needed to reach the criterion of 95% 'correct' trials (trial error <1) over 100 successive trials (843 trials); dotted vertical lines indicate the inter-quartile range (692–1125 trials). Performance (i.e., magnitude of the response error) continues to improve after reaching criterion and reaches a low, stable residual asymptote. ( (bottom right): Activities of 6 different neurons, including the output neuron (thick black line), for two stimulus combinations, before training (left) and after training (right). Note that neural traces remain highly dynamical even after training.

different (AB or BA), in which case the network should output 1 (see Materials and methods for a detailed description).

This simple task exhibits several interesting features. First, it is arguably the simplest possible flexible (that is, context-dependent) decision task: on sensing the second stimulus, the network must produce a different response depending on the identity of the first stimulus. Second, because the intervening delay is much longer than the neural time constant ($\tau$ = 30 ms, see Materials and methods), the network must maintain some memory of the first stimulus before the second stimulus arises. Third, to solve this task, some neurons in the network must necessarily

possess some form of nonlinear mixed selectivity (note that the problem is in essence a delayed exclusive-or problem), a hallmark of neural activities in prefrontal cortices (*Rigotti et al., 2013*).

The networks consistently learn to perform the task with high accuracy. *Figure 1b* shows the time course of the median error over 20 training runs, each starting with a different randomly initialized network. The 'error' for a trial is the mean *absolute* difference between the output neuron's response and the correct response, i.e. 1 or −1, over the last 200 ms of the trial (see Materials and methods). Shaded area indicates 1 st and third quartile over the 20 runs. To define a measure of successful performance, we set a criterion of 95 trials with error lower than 1 (indicating the mean response is closer to the correct than to the incorrect response, i.e. of the correct sign) over 100 successive trials ($p<10^{-20}$ under random choice, binomial test). The median time to criterion across 20 runs is 843 trials (inter-quartile range: 692–1125). Response error reliably converges towards a very low residual value.

How does the network represent and maintain traces of incoming stimuli? One possibility is that certain neurons encode stimulus identity by maintaining a stable 'register' value over time, such that the firing rate of certain cells directly specify stimulus identity in a relatively time-independent manner. By contrast, physiological studies suggest that neural coding during a working memory task is highly dynamic, with stimulus identity being represented by widely fluctuating patterns of neural responses, in such a way that the tuning of individual neurons significantly changes over the course of a trial (*Meyers et al., 2008*; *Barak et al., 2010*; *Stokes et al., 2013*). As shown in *Figure 1c*, trained networks do exhibit highly dynamic responses over the course of a trial. This suggests that the networks might make use of dynamic representations.

To analyze the encoding and maintenance of stimulus identity over time in the network, we used a cross-temporal classification approach (*Meyers et al., 2008*; *Stokes et al., 2013*; *Dehaene and King 2016*). We trained a maximum-correlation classifier to decode various task-relevant features (identity of first and second stimulus, and final response), based on whole-population activity at any given time, and then used these time-specific classifiers to try and extract the same task-relevant features at all possible points in time. This method can detect not only whether the network encodes a certain task-relevant variable, but also whether the representation of this variable changes over time (see Materials and methods).

The results in *Figure 2* suggest a highly dynamic representation of stimuli by the network. For example, the identity of the first stimulus can be successfully decoded during both first and second

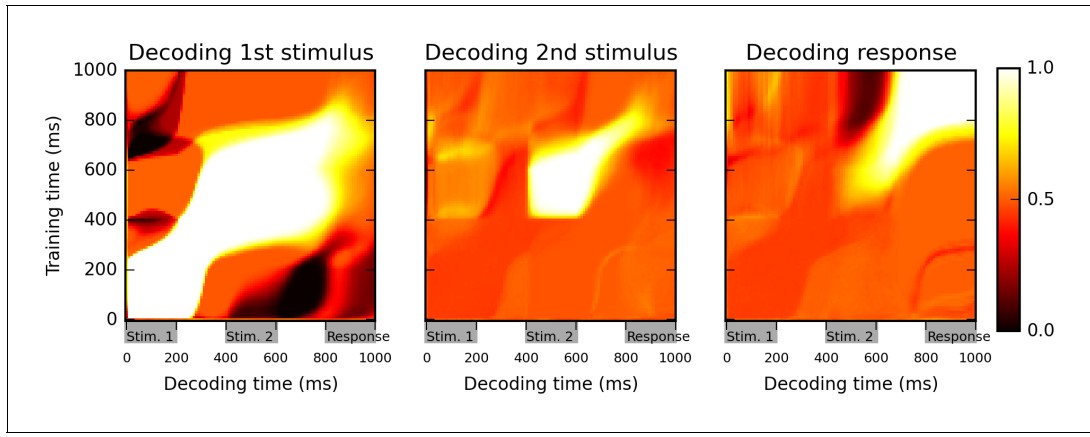

**Figure 2.** Cross-temporal classification performance reveals dynamic coding. Cross-temporal classification of 1st stimulus identity (left panel), 2nd stimulus identity (middle panel) and network response (right panel). Row i and column j of each matrix indicates the accuracy of a classifier, trained on population activity data at time i, in guessing a specific task feature using population activity data at time j (training and decoding data are always separate). While the network reliably encodes information about stimulus identity right until the onset of the response period (as shown by high accuracy values along the diagonal in left and middle panel), this information is stored with a highly dynamic encoding (as shown by low cross-classification accuracy across successive periods, i.e., 'bottlenecks' with high accuracy on the diagonal but low accuracy away from the diagonal). Note that in both left and middle panels, stimulus identity information decreases greatly at the onset of the response period, reflecting a shift to a from stimulus-specific to response-specific encoding (see also *Figure 3*).

stimulus presentations, as well as during the intervening delay, as shown by high classification accuracy values on the diagonal during this entire period (*Figure 2*, left panel). However, the cross-temporal classification performance between these two periods, as seen on the off-diagonal areas (for example, in the areas at 0–200 ms on one axis and 400–600 on the other, corresponding to training the classifier based on data from one stimulus presentation and testing it on data from the other stimulus presentation) is essentially at chance level (accuracy ~0.5), or even below chance (dark patches). This suggests that while the network reliably encodes information about 1st-stimulus identity across the first 800 ms of the trial, the way in which this identity is represented changes widely between successive periods within the trial. Similarly, the 2nd-stimulus identity is maintained from its onset until the beginning of the response period, but in a dynamical manner (low off-diagonal, cross-temporal accuracy between the 400–600 ms period and the 600–800 ms period, in comparison to the high diagonal accuracy over the entire 400–800 ms period).

Another feature of these plots is that the accuracy of stimulus identity decoding strongly decreases over the course of the 'response' period (low values along the diagonal for the 800–1000 ms in first and second panel of *Figure 2*). This suggests that the network largely stops maintaining information about the specific identity of previous stimuli, and instead encodes solely the actual response, as shown by the very strong classification accuracy in the upper-right portion of the third panel.

To test this interpretation, following (*Stokes et al., 2013*), we produce Multi-dimensional scaling (MDS) plots of population activity at different time points and for different stimulus conditions (*Figure 3*). MDS attempts to find a two-dimensional projection such that the distance between any two data points is as similar as possible to their actual distance in the full-dimensional space: nearby

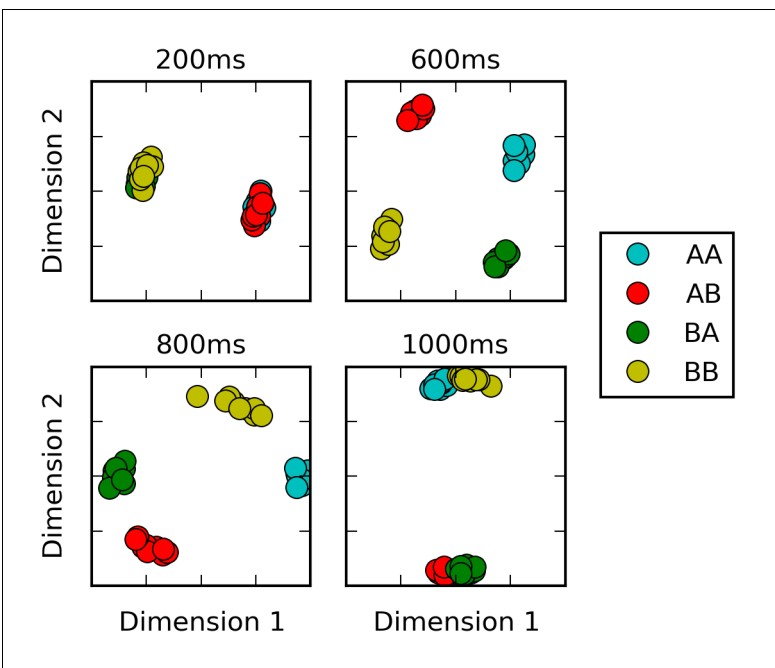

**Figure 3.** Multi-dimensional scaling plots of population activity reflect shifting encodings of task-relevant information. Population response vectors at various points in time (color-coded by stimulus combination) are projected in two dimensions while preserving distances between data points as much as possible, using multi-dimensional scaling. At the end of the first stimulus presentation (200 ms), population states are firmly separated by first stimulus identity, as expected. After the second stimulus presentation (600 ms), all four possible stimulus combinations lead to clearly separate population activity states. However, population states corresponding to different responses start to cluster together at the onset of the response period (800 ms). Late in the response period (1000 ms), population trajectories corresponding to the same response (AA and BB, or BA and AB) have largely merged together, reflecting a shift from stimulus-specific to response-specific representation and a successful 'routing' of individual stimulus-specific states to the adequate response-specific state.

(distant) population states should thus produce nearby (distant) points on the MDS plot. Early in the trial, all possible stimulus identity combinations generate different, consistent trajectories, indicating stimulus-dependent encoding. By the late response period, however (1000 ms), the trajectories have essentially merged into two clusters, corresponding to the network response ('same' or 'different') and largely erasing any distinction based on the specific identity of either first or second stimulus. Thus, during the response period, the network moves from a stimulus-specific representation to a response-specific representation: the stimulus-specific response is flexibly routed to the appropriate, context-dependent response state, as previously observed in cortical activity during a flexible association task (*Stokes et al., 2013*).

## Task 2: Flexible selective integration of sensory inputs

An important aspect of cognitive control is the ability to attend selectively to specific portions of the sensory input, while ignoring the rest, in a flexible manner. Recently Mante, Sussillo and colleagues have studied the neural basis of this ability in macaque monkey prefrontal cortex (*Mante et al., 2013*). Monkeys were trained to report either the dominant color or the dominant motion direction of randomly-moving colored dots. Thus, the same stimulus could entail different appropriate responses depending on current context (i.e. which modality - color or motion - was relevant for this trial). Furthermore, due to the noisy stimulus, the task required selective temporal integration of the relevant sensory input. In addition to neural recordings, Mante and colleagues also trained a recurrent neural network to perform the same task, using supervised learning based on Hessian-free optimization. By analyzing the trained network, they identified mechanisms for selective integration of context-dependent inputs in a single network (*Mante et al., 2013*). This task was also used as an example application by Song and colleagues for their recurrent network training framework (*Song et al., 2016*).

We trained a network to perform the same task, using our proposed plasticity rule (see *Figure 4*). Our settings are deliberately similar to those described by Mante, Sussillo and colleagues. The network has two 'sensory' inputs (representing the two stimulus modalities of motion and color), implemented as random (Gaussian) time series, with a randomly chosen mean for each trial; the mean of each time-series thus represent the 'value' of the corresponding modality for this trial. In addition, two 'context' inputs specify which modality is relevant for each trial. The network must produce output 1 if the context-indicated sensory input has a positive mean, or −1 if it has negative mean (see Materials and methods for a detailed description).

*Figure 4b* shows the psychometric curves of a fully-trained network, that is, the mean response as a function of stimulus value. For either modality, we show separate psychometric curves for when this modality was the relevant one and when it was irrelevant. When trials are sorted according to the value of the relevant modality, responses form a steep sigmoid curve with a relatively sharp transition between −1 and +1 centered roughly at 0. By contrast, when trials are sorted according to the value of the irrelevant modality, responses are evenly distributed across the entire range. Thus, the network accurately responds to the relevant signal, while largely ignoring the irrelevant one in each context (Compare to Figure Extended Data 2 in *Mante et al., 2013*). This indicates that the network has learned not only to perform temporal integration of an ambiguous, stochastic input, but also to flexibly 'attend' to different input streams depending on context.

How is information represented in the network over time? We use Mante and Sussillo's orthogonal decoding procedure, which seeks to extract independent measures of how various task features (stimulus values, context, decision) are encoded in the network (see Materials and methods). Briefly, this method consists in using multiple linear regression to measure how much certain task-relevant features are being *independently* represented by the network at any time (see Materials and methods for a detailed description). The results are shown in *Figure 5* (compare to *Figures 2* and *5* in *Mante et al., 2013*). These trajectories plot the evolution of network information over time, for various combinations of context (relevant modality) and task features. Each trajectory represents the average population activity, at successive points in time, of all correct trials that have the same bias value for the averaging modality and resulted in the same final choice. Trajectories are colored according to the bias value of the averaging modality, ranging from −0.25 (bright red) to 0.25 (bright green). We project these averaged population trajectories along the orthogonal feature dimensions extracted by orthogonal decoding, which tells us how strongly the network encodes this

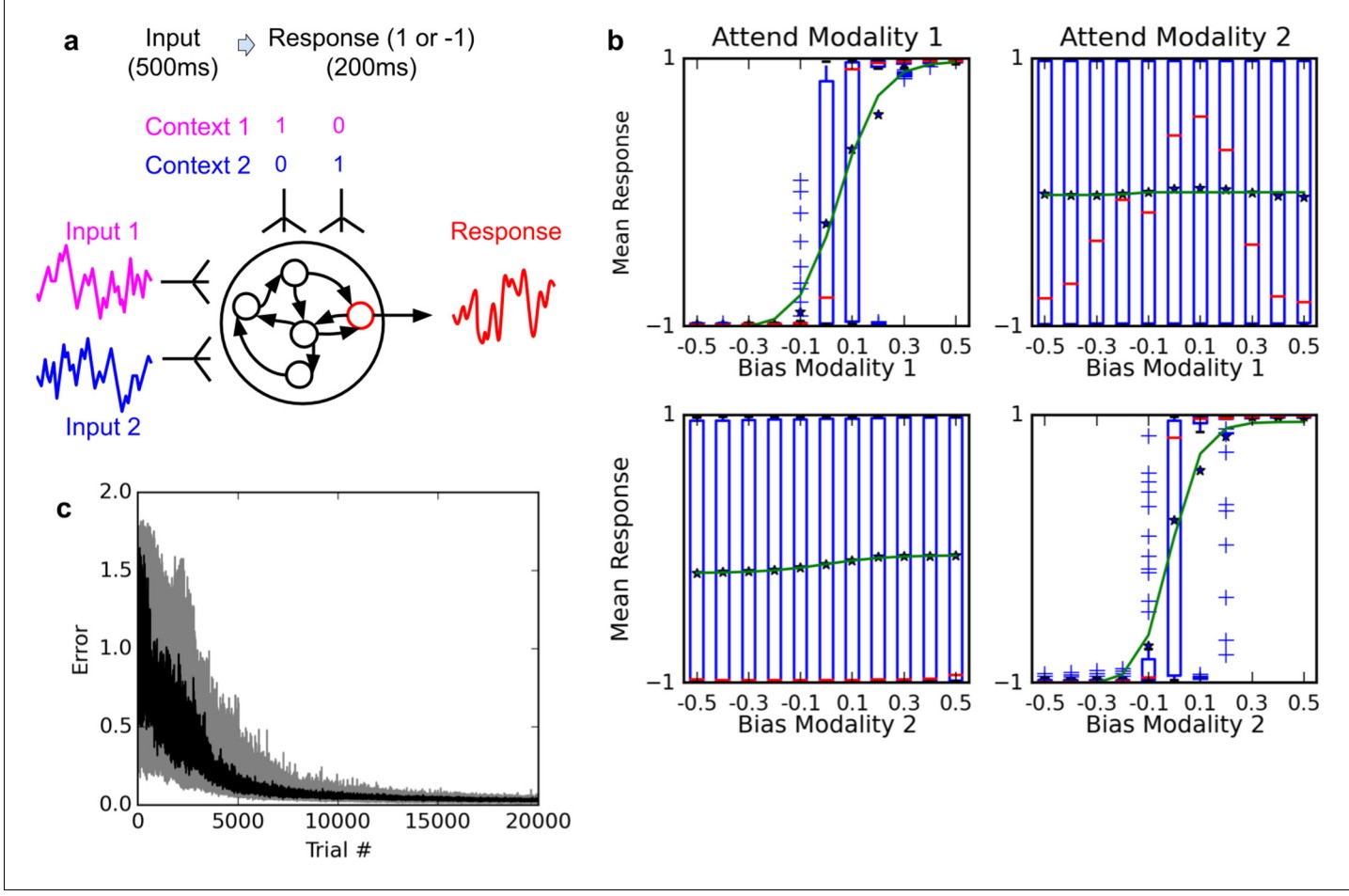

**Figure 4.** Selective integration task. (**A**) (top left): task description. The network receives two noisy inputs, simulating sensory information coming from two different modalities, as well as two 'context' inputs to indicate which of the two sensory inputs must be attended. The task is to produce output 1 if the cued input has a positive mean, and −1 if the cued input has negative mean; this task requires both selective attention and temporal integration of the attended input. (**B**) (top right): Psychometric curves of network responses. Responses are segregated according to the value of the relevant modality bias, and shown as box-plots: blue boxes indicate the inter-quartile range, with data points outside the box showing as blue crosses; red bars indicate medians, and dark stars indicate means; green curves are sigmoid fits to the means. Top-left panel: Responses when context requires attending to modality 1, sorted by the bias of modality 1 inputs. The network response correctly tracks the overall bias of modality 1 inputs. Bottom-left panel: same data, but sorted by modality 2 bias. Network response is mostly unaffected by modality 2 bias, as expected since the network is required to attend to modality 1 only. Right panels: network responses when the context requires attending to modality 2. Again, the network correctly identifies the direction of the relevant modality while mostly ignoring the irrelevant modality. (**C**) (Bottom): Median and inter-quartile range of trial error (mean absolute difference between output and correct response over the last 200 ms of the trial) over 20 runs of 20000 trials each.

particular feature at a given time. We then plot the resulting trajectories in feature dimension sub-spaces ('final choice' dimension is always used as the horizontal axis, while the vertical axis may be either of the two sensory modality dimensions).

As observed in cortical recordings (*Mante et al., 2013*), these trajectories reveal that *both* the relevant and the irrelevant modality are actually represented in the network: the trajectories for varying value of either modality form an ordered progression in the corresponding 'modality' dimension (y-axis), even when that modality is irrelevant (bottom-left and top-right panels in *Figure 5*); however, only the relevant modality correlates with representation of final choice (compare panels where trajectories are separated by value of the relevant vs. irrelevant modality), in accordance with physiological observations (*Mante et al., 2013*). This confirms that the network has learnt to selectively integrate the context-indicated variable while discarding the irrelevant one for each trial.

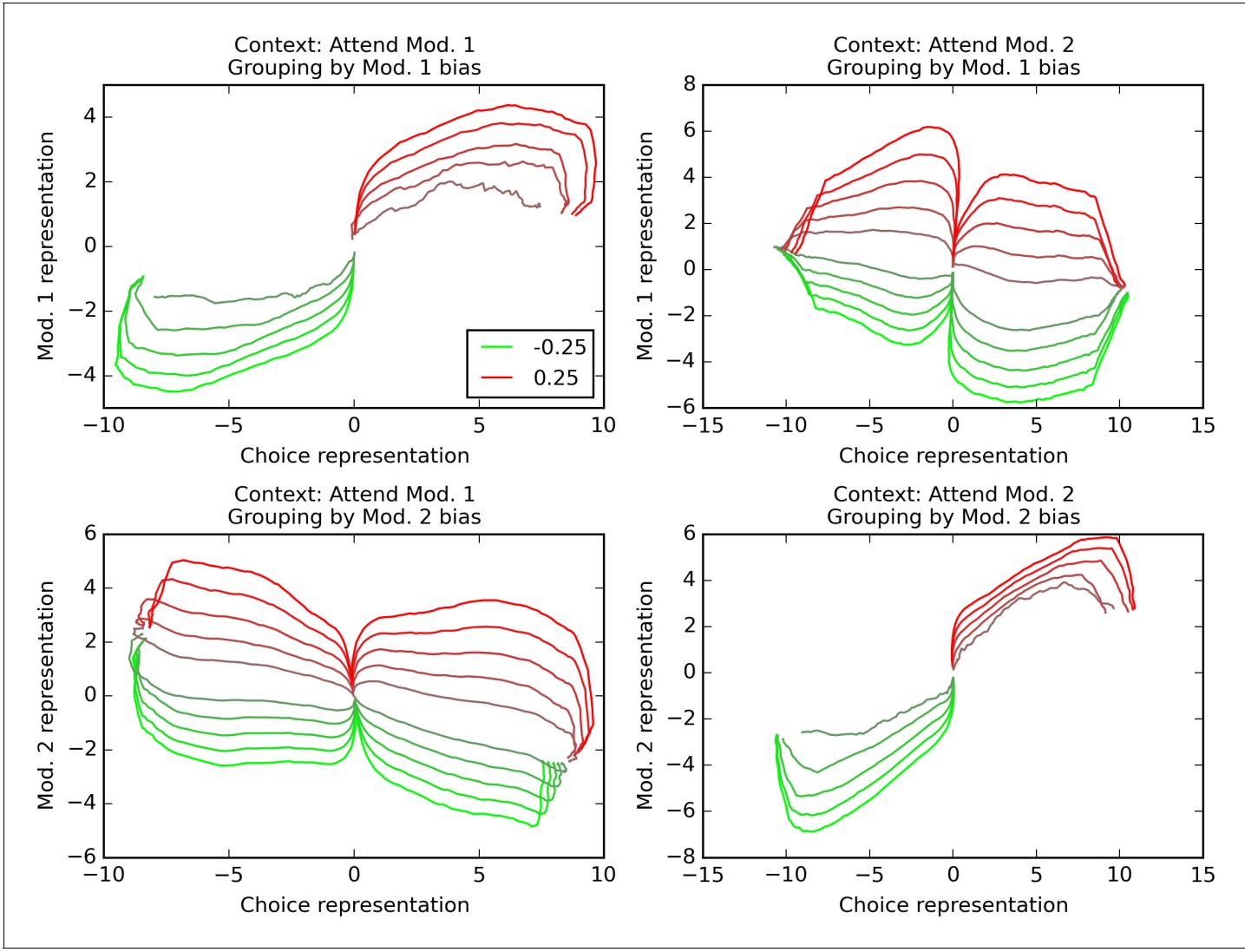

**Figure 5.** Orthogonal decoding of population activities. Population response patterns are averaged at each point in time, separately by context (i.e. relevant modality), final choice of the trial, and the bias of modality 1 (top) or 2 (bottom). For each combination of relevant modality and averaging modality, these averaged patterns over time result in different trajectories (one per value of the averaging modality, ranging from −0.25 (bright red) to 0.25 (bright green)). We project these trajectories over dimensions indicating how reliably the network encodes modality 1 value, modality 2 value, and final choice. In all graphs, the x axis is the dimension that reflects current encoding of final choice; the y axis is the dimension that reflects current encoding of the grouping modality (i.e. the one used for averaging). Only correct trials are used (thus top-left and bottom-right panels only have 10 trajectories, since correct trials with positive bias in the relevant modality cannot lead to a negative choice, and vice versa). The trajectories reveal that the network encodes both the relevant and the irrelevant modality, though only the relevant one is linked to the final choice. See text for details.

## Task 3: Controlling a musculoskeletal model of the human arm

In both of the previous tasks, the network output was a single response channel. However, flexible behavior often requires coordinating multiple outputs, especially during movement. To test whether our plasticity rule can produce coordinated multiplexed responses, we trained a network to control a biomechanical model of the human arm. The model is a custom modification of the one described in *Saul et al. (2015)* (itself an extension of *Holzbaur et al., 2005*) and uses the Thelen muscle model (*Thelen, 2003*). The model implements the human upper skeleton, with 4 degrees of freedom (three at the shoulder, one at the elbow), actuated by 16 muscles attached to the shoulder, chest, and upper and lower arm bones. Each of the 16 muscles is controlled by a specific network output cell. The task is to reach towards one of two spherical targets, located in front of the body on either side

of the sagittal plane. The appropriate target ball for each trial is indicated by two input channels, set either to 1 and 0 (left-side target) or to 0 and 1 (right-side target) respectively for the entire duration of the trial (700 ms). No other inputs are provided to the system. The error at the end of each trial is measured by the absolute distance between the tip of the hand and the center of the target ball, plus a small penalty for total muscle activation over the entire trial. Note that while the target balls are symmetrically arranged with regard to the body, they are not symmetrical with regard to the right arm (which is the one we model): the right-side ball is closer than the left-side one, and thus reaching either target requires qualitatively different movements.

Results are shown in *Figure 6*. Initially, as expected, the untrained network performs random, aimless movements, resulting in high initial error (*Figure 6b*). Performance improves almost from the start of the training process, reaching a low residual error after about 3000 trials. To visualize the impact of training on the dynamics of population activity, we project the population activity over its

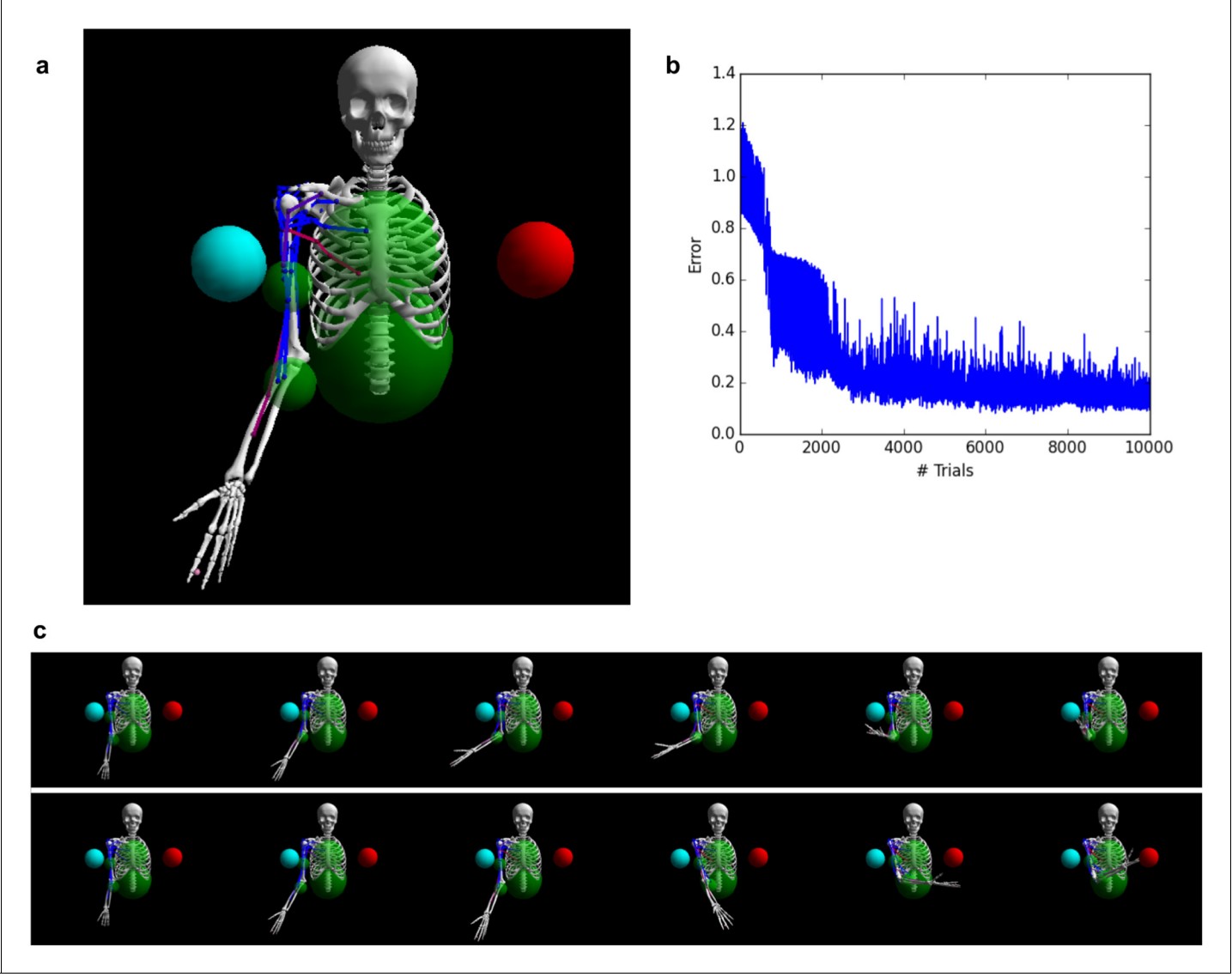

**Figure 6.** Controlling a biophysical model of the human arm. (**A**) (top left): A model of the human upper skeleton with 4 degrees of freedom (shoulder and elbow joints), actuated by 16 muscles at the shoulder, chest and arm (colored strings represent muscles; blue indicates low activation, red indicates high activation). The task is to reach either of two target balls (blue or red), depending on a context input. (**B**) (top right): During training, the error (measured by the distance between tip of hand and target ball at the end of each trial) improves immediately and reaches a low residual plateau after about 3000 trials. (**C**) (bottom): frame-by-frame illustrations of a right-target trial (top row) and a left-target trial (bottom row), after training.

first three principal components at successive points in time over the course of each trial, using 16 trials for either target context, both before and after training (i.e., 64 trials in total). The fully trained network correctly reaches the adequate target according to context (*Figure 6c*).

## Additional experiments: long delays, variable timing, and excitatory-inhibitory networks

Here we return to the delayed nonmatch-to-sample task, in order to test the learning rule under various changes in experimental conditions. All results in this section are obtained with the delayed nonmatch-to-sample task as described above, with modifications specified below.

First, we test the ability of the rule to deal with much longer delays. We extend the stimulus-absent delay period, from 200 ms to one full second. In addition, stimulus presentations are extended from 200 ms to 400 ms. With the 200 ms response period, this sums up to a total time of 2000 ms for each trial. We found it useful to reduce the learning rate from 0.1 to 0.03. As shown in *Figure 7a*, the rule still learns the task reliably, although reaching criterion (95% of trials with a mean absolute error over the response period lower than 1) requires more trials (median over 20 runs: 2230 trials, inter-quartile range:1366–4573 trials).

We then test the ability of the rule to deal with variable trial structure, and in particular, with variable stimulus timing. For each trial, we randomly pick the inter-stimulus delay period between 300 and 800 ms (as opposed to the previous fixed 200 ms duration), while stimulus presentation is extended to 300 ms. The total duration of the trial is always 1600 ms, with the last 200 ms being the response period. As shown in *Figure 7b*, the rule still manages to learn the task; however, the number of trials required to learn the task is now much larger. Indeed, the increased variance introduced by variable stimulus timing had a strong destabilizing effect: we found it necessary to reduce the learning rate to 0.003 to obtain reliable learning. Thus, variable stimulus timing makes learning much more difficult, though still feasible.

Finally, we modify the network model to make it more realistic, by enforcing nonnegative neural responses and separate populations of strictly excitatory and strictly inhibitory neurons (Dale's law). In accordance with most related work (*Sussillo and Abbott, 2009*; *Hoerzer et al., 2014*; *Sussillo et al., 2015*), previous experiments used a canonical, widely studied recurrent network model (*Sompolinsky et al., 1988*), where neural responses can be negative and neurons send both positive and negative connections (see Materials and methods). However, recently, Mastrogiuseppe and Ostojic (*Mastrogiuseppe and Ostojic, 2016*) showed that strong theoretical results could be extended to networks with nonnegative responses and separate populations of strictly excitatory and inhibitory neurons; in particular, they proved the existence of a connectivity regime that guarantees ongoing, chaotic activity in the network, without reaching saturation.

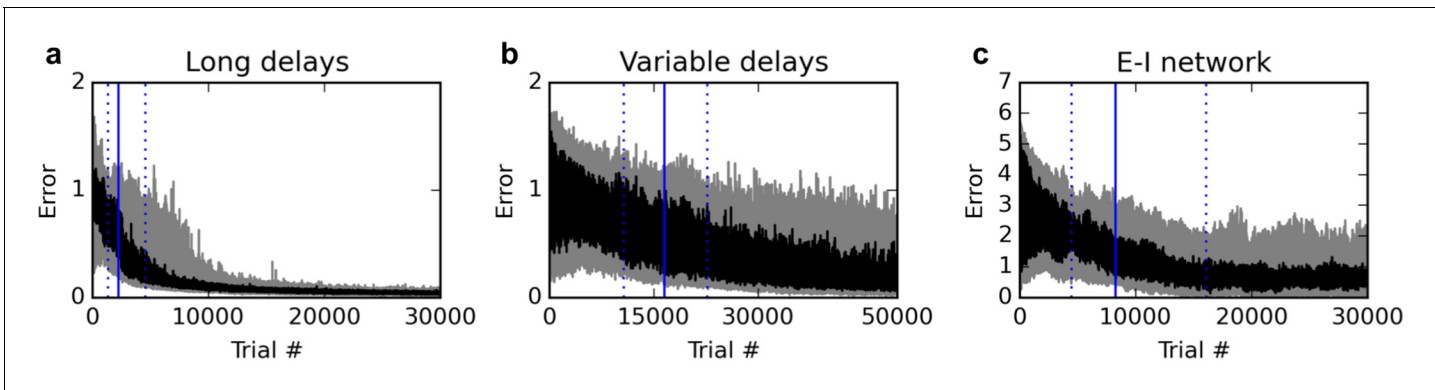

**Figure 7.** Additional experiments. Panels show learning curves for the delayed nonmatch-to-sample task, modified in three ways. (A) (left): long 1000 ms delays. (B) (middle): variable inter-stimulus interval, randomly chosen from 300 to 800 ms. (C) (right): Networks with nonnegative neural responses and separate excitatory and inhibitory neurons (in accordance with Dale's law). Conventions are as in *Figure 1*: dark lines indicate median over 20 runs, while gray shaded area indicates the inter-quartile range. See text for details.

We sought to test whether the rule proposed here can be used to learn cognitive tasks in a network of this type. We implemented a network with separate excitatory and inhibitory neurons, initialized with semi-sparse connections, with nonnegative, piecewise-linear activation functions (see Materials and methods). Importantly, the responses are now constrained between 0 and 20 (although in practice they never exceed ~10, thus remaining far from saturation). The target response for identical stimuli is 0 (instead of −1), and the target response for different stimuli is 5 (instead of 1). The criterion is now to reach 95% of (absolute) errors below 2.5, which ensures that responses are closer to the correct than the incorrect response.

As shown in *Figure 7c*, the rule can still reliably learn the task with this more realistic network. While learning takes longer, a direct comparison is difficult due to the differences in network activity and the wider range of possible responses. Nevertheless, these results confirm that the proposed rule is applicable to more realistic networks with nonnegative, non-saturating responses and separate excitatory and inhibitory neurons.

## Analysis of the proposed rule

Here we attempt to reach a better understanding on how and why the learning rule actually works. We also put the rule in the context of related learning algorithms, including the node-perturbation method (*Fiete et al., 2006*) and the Exploratory Hebbian method (*Hoerzer et al., 2014*; *Legenstein et al., 2010*). We support our discussion with simple quantitative experiments that allow us to isolate the impact of various factors on the performance of the learning rule.

### Overview

The rule proposed here is a more biologically plausible implementation of the so-called node-perturbation rule (*Fiete et al., 2006*, *2007*), which is itself a variant of the classical REINFORCE algorithm (*Williams, 1992*). Node-perturbation (like REINFORCE) consists in applying random perturbations to neural responses, then modifying the weights so as to make future responses more similar (resp. less similar) to the perturbation-induced responses, if the perturbed outputs led to a better-than-expected (resp. worse-than-expected) reward. Like the Exploratory-Hebbian (E-H) method of Legenstein, Hoerzer and Maass (*Legenstein et al., 2010*; *Hoerzer et al., 2014*), our rule extracts exploratory perturbations from outputs by subtracting a running average from ongoing neural responses. Unlike the E-H method, however, it can actually learn from sparse, delayed rewards at the end of each trial, without requiring a continuous, real-time reward signal. As we explain below, this is due to the supralinear amplification of plasticity increments.

### Node-perturbation and REINFORCE

The node-perturbation rule (*Fiete et al., 2006*, *2007*) is a reinforcement learning rule that can train neural networks using only sparse, delayed reward signals to guide the learning. The intuitive mechanism of the rule is to apply random exploratory perturbations to neural output, then modify the weights so that future, unperturbed outputs will be more similar to this perturbed output if this perturbed output turned out to elicit a 'good' reward - or conversely, less similar if it produced a 'poor' reward (note that this is the central idea of the REINFORCE algorithm [*Williams, 1992*]).

The node-perturbation method can be summarized as follows (*Fiete et al., 2006*):

1- During each episode, apply random exploratory perturbations $\xi(t)$ to the neuron's output $y_i(t)$.

2- At each synapse, compute the so-called *eligibility trace* $e_{i,j}$ by accumulating the products of output perturbations by current input at this synapse at the time of perturbation: $e_{i,j} = \Sigma_t \xi_i(t) x_j(t)$

3- At the end of each episode, compute the reward R for this episode.

4- Add to each synaptic weight the accumulated eligibility trace, multiplied by the net (baseline-subtracted) reward for this episode: $\Delta w_{i,j} = \eta * e_{i,j} * (R - R_0)$.

In step 4, $\eta$ is a learning rate parameter, and $R_0$ is the predicted reward for this episode in the absence of perturbations; thus, $R - R_0$ reflects whether the trajectory of outputs produced during this episode was 'better' or 'worse' than usual.

The effect of step 4 is to make future trajectories more similar to the just-experienced perturbed trajectory (in response to the same inputs), if this perturbed trajectory produced a 'good' reward; or less similar, if the trajectory produced a 'bad' reward. This is because adding $\xi(t)x_j(t)$ to $w_{i,j}$ will move future responses to the same input $x_j(t)$ in the direction of $\xi(t)$, and thus tend to reproduce the

stochastically perturbed response. This can be easily shown analytically (notice that $\frac{\partial y(t)}{\partial w} = x(t)$, at least in the limit of a linear neuron). It also makes intuitive sense: if the perturbation was positive (resp. negative), this rule would add a positive (resp. negative) multiple of x(t) to the synaptic weight, which would make the weights more (resp. less) correlated with input x(t) and thus produce a higher (resp. lower) response to future presentations of the same input x(t).

While node perturbation was introduced by Fiete and Seung (*Fiete et al., 2006*), the original REINFORCE paper describes several rules that implement node-perturbation, i.e. modifying weights by the accumulated product of inputs by perturbations, multiplied by net rewards (see e.g. Eq. 11 in *Williams, 1992*).

## The E-H method: Hebbian implementation of node-perturbation by subtracting running averages

A difficulty with the node-perturbation rule is that it is hard to reconcile with existing models of synaptic plasticity. It requires that each synapse maintains a distinction between the 'actual' inputs and the 'perturbation' input, and then perform learning by multiplying these two different types of inputs. The biological mechanism for such learning is not obvious. This is also in contrast with standard models of plasticity based on Hebbian learning, that is, a product between inputs and outputs rather than between two forms of input.

Legenstein, Hoerzer and Maass have proposed a method, which they call the Exploratory Hebbian (E-H) method, to implement node-perturbation in a biologically plausible, Hebbian manner, using information local to the synapse (*Legenstein et al., 2010*). The central idea is that if perturbations are fast and strong enough, they can be extracted from ongoing neuron output by subtracting a fast running average, which should isolate fast fluctuations in neural responses. The eligibility trace is then computed as the product of inputs by these fast fluctuations in output (which are deemed to represent mostly the exploratory perturbations), producing a Hebbian, synapse-local rule which implements the node-perturbation rule.

The E-H rule is expressed formally as follows (*Legenstein et al., 2010*):

$$\Delta(w_{i,j}(t) = x_j(t)(y_i(t) - \bar{y}_i(t))(R(t) - \bar{R}(t))$$

Where $x_j$ is the input at the synapse, $y_i$ is the neuron's output, and the overbar denotes a short-term running average (i.e. $\bar{Y}(t + 1) = \alpha\,\bar{Y}(t) + (1-\alpha)\,y(t)$ where $\alpha$ is a constant). Thus, $y_i(t)$ - $\bar{Y}_i(t)$ extracts fast fluctuations in neural output, while R(t) -$\bar{R}(t)$ extracts fast fluctuations in the reward signal.

Hoerzer and colleagues (*Hoerzer et al., 2014*) showed that this rule still works if the R(t)-$\bar{R}(t)$ term is replaced by a less informative quantity M(t) which is 1 if R(t)>$\bar{R}(t)$ and 0 otherwise. They also showed that this rule can be used to successfully train chaotic recurrent networks for complex tasks.

## Removing the need for continuous reward signals with supralinear amplification

A difficulty with the E-H rule is that it requires a continuous, real-time reward signal R(t): at every point in time, the system must know whether it is doing better or worse than before. This negates a central advantage of reinforcement learning: the ability to learn from sparse, delayed rewards.

One reason why real-time reward signals are needed in the E-H rule is that simply subtracting a running average from ongoing activity does not reliably isolate external perturbations, due to spurious relaxation effects. To take a maximally simplified example, consider the example trace in *Figure 8*. This represents the output of a single neuron receiving constant inputs, with a single positive perturbation at time T = 100 (top panel). To extract fluctuations, we subtract a running average from the output (bottom panel).

When the perturbation is applied, initially the output is larger than the running average, and thus the difference reflects the perturbation ('Perturbation effect' gray area). However, the running average then increases to include the recent perturbed outputs, and now the difference between decaying ongoing activity and running average switches to *negative* ('Relaxation effect' gray area). These negative relaxation terms will be accumulated into the Hebbian product and counteract the positive, perturbation-related initial terms. In fact, for the simple case shown in *Figure 8* (constant input and

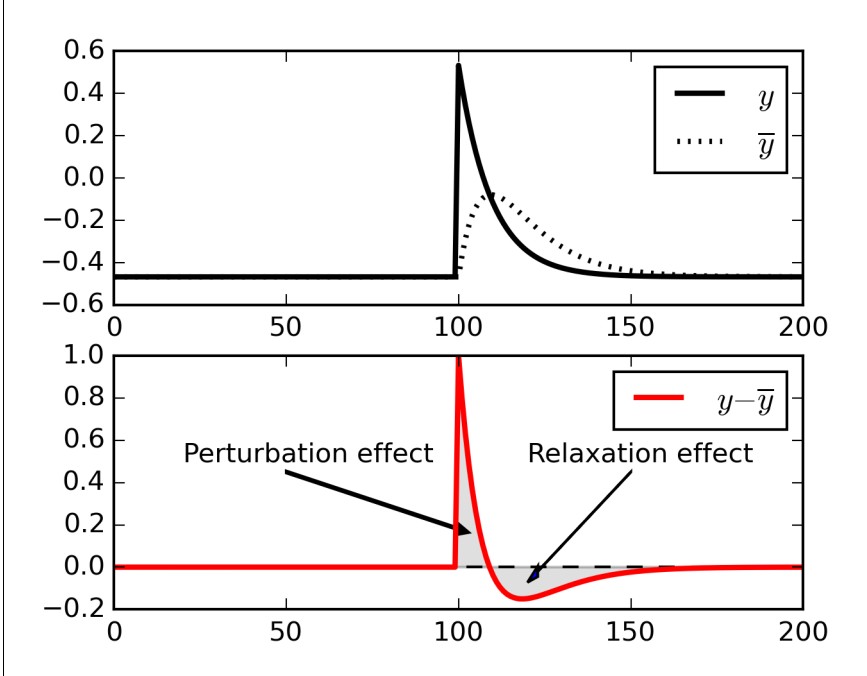

**Figure 8.** Relaxation effects. When a perturbation is applied to signal, subtracting a running average initially extracts the perturbation, but then introduces opposite-sign terms as the running average relaxes to the signal.

a single perturbation) the total sum of all $y(t) - \bar{y}(t)$ terms over a full episode will converge to zero, and so will the eligibility trace (being a product of constant inputs by fluctuations summing to zero).

In more complex cases such as recurrent networks considered here, with time-varying inputs, the cancellation will only be partial, depending on the amount of autocorrelation in the inputs. The eligibility trace will include both the product of the 'perturbation effect' by the inputs at the time of the perturbation (which is what we want), and also the product of immediately subsequent, unpredictable inputs by the 'relaxation effect' (which is undesirable). The latter part adds uncontrollable ('garbage') components to the eligibility trace. Furthermore, if there is any temporal correlation in successive inputs, then these 'garbage' components will tend to partially cancel out the perturbation-related terms. This, we suggest, causes the inability of the E-H rule to learn without continuous, real-time reward signal (we show experiments in support of our interpretation below, in *Figure 9*).

The Exploratory Hebbian method negates this undesirable effect by using a real-time reward signal $R(t)$ which is also high-passed by subtracting its own running average $\bar{R}(t)$. Now the same relaxation effects will occur both in the $y$ and the $R$ traces, and thus the product of relaxation terms in both traces results in a positive value, which will actually reinforce the perturbation-related terms rather than cancel them (assuming that inputs have any significant temporal autocorrelation).

This analysis immediately suggests an alternative solution to the problem. Notice that the perturbation-related fluctuation is large, while the subsequent countering relaxation terms are small ('Perturbation effect' vs. 'Relaxation effect' in *Figure 8*). Thus, if we impose a *supralinear* function on plasticity increments, the large perturbation-related terms will be amplified, while the small relaxation-related terms will be suppressed. This leads to the plasticity rule proposed in the present paper.

Note that this is conceptually related to recently-proposed thresholded Hebbian rules, whereby plasticity is only triggered by events in which the Hebbian product reaches a certain threshold (*Soltoggio and Steil, 2013*). A supralinear amplification offers a smoother amplification of larger Hebbian events, by comparison to the all-or-nothing effect of a threshold; however, the overall effect is similar: ignore small, possibly incidental correlations of input and output, but retain the larger ones, which are more likely to be informative.

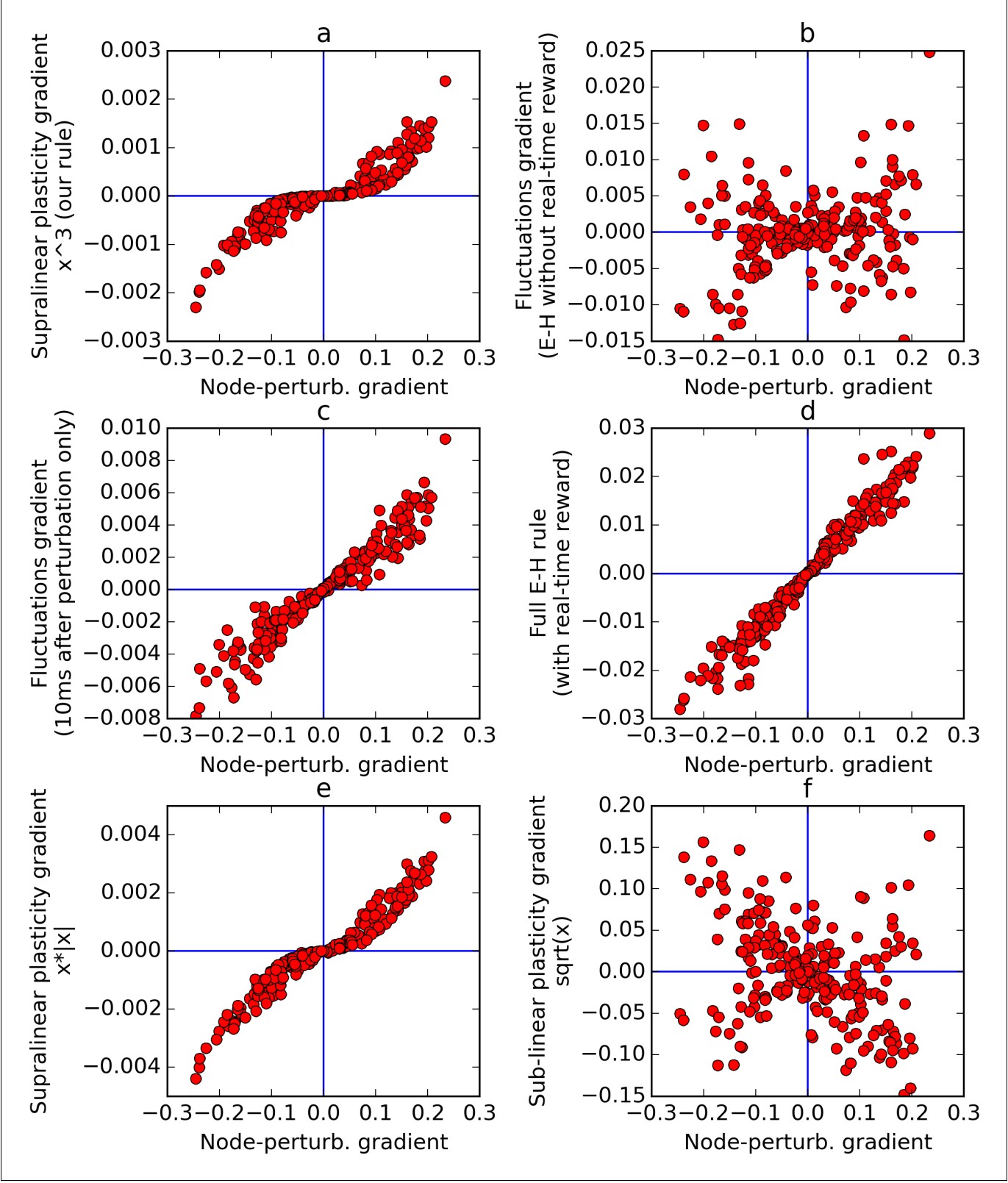

**Figure 9.** Comparison of error gradients. A recurrent network is repeatedly exposed to randomly-chosen, time-constant inputs, and must learn to determine whether the inputs have positive sum. We compute the learning gradients over the weights according to various methods, for many trials, based on a single perturbation at a fixed time in each trial. In all four panels, the x-axis indicates the gradient computed by node-perturbation, used as a ground truth. Panel **a**: the weight modifications produced by node-perturbation align remarkably with the rule described in this paper. Panel **b**:

*Figure 9 continued on next page*

*Figure 9 continued*
gradients computed by using raw fluctuations of output about a running average, without supralinear amplification, are essentially random. Panel **c**: if we restrict the plasticity computations to the first 10 ms after perturbation, the correct gradients are recovered (using only 1 ms would be identical to panel **a**), confirming that post-perturbation effects are responsible. Panel **d**: The full E-H rule, with real-time reward signal, also recovers the node-perturbation gradients. Panel **e**: Using a different supralinear function (signed square rather than cubic) produces largely similar results to Panel **a**. Panel **f**: By contrast, a sublinear function (square root) results in largely random gradients.

## Why not just use reward-modulated hebbian learning?

Node-perturbation increases weights by a product of (net) rewards, inputs and perturbations of outputs. Why use the perturbations, rather than the raw outputs, in the product? Why not just use the standard Hebbian product of inputs by outputs (rather than perturbations), and multiply this by the reward? This is because the full Hebbian products (inputs-outputs) are dominated by reward-unrelated terms. The Hebbian products tend to simply reinforce all existing co-occurrences of inputs and outputs, regardless of how this increase would affect performance. By contrast, the perturbation-based products ensure that the weights are modified specifically to reproduce the perturbation-caused change in trajectory. The two have no reason to correlate, and the former is usually larger than the latter if perturbations are sparse.

In theory, it is possible to average out the non-reward-related component of reward-modulated Hebbian learning, by making sure that the reward signal is absolutely zero-centered separately for each trial type (that is, for each possible combination of input range and target outputs) (*Frémaux et al., 2010*; *Sprekeler et al., 2009*). This corresponds to the role of a *critic* (a complex reward predictor) in reinforcement learning, and is implemented in the present paper by maintaining separate values of $\bar{R}$ for each trial type. However, this procedure leaves us with the problem of considerable variance, which suffices to make learning all but impossible in complex tasks (see the specific discussion of this point in *Legenstein et al. (2010)*: as connection weights increases, the purely-Hebbian component will tend to take over and drown the reward-related component).

This problem is compounded in recurrent networks such as the one described here, because now a single perturbation can have significant, unpredictable effects on future neural responses over the rest of the trial. For example, a single *positive* perturbation can have an arbitrary indirect effect on the response of the perturbed neuron, even making it eventually *lower* than it would have been over an extended period of time. As a result, the accumulated product of inputs and outputs would not be dominated by the (single-point) response increase caused by the perturbation, but by the (extended) subsequent decrease in response, which would actually have unpredictable effects on weight modification and future trajectories. Note that this effect (Hebbian learning futilely learning the co-activations during the perturbed trajectories, rather than the perturbations that cause the changes in trajectories) cannot be averaged out by centering the rewards, since the trajectories are what determines the rewards.

By contrast, node-perturbation only modifies weights to incorporate products of perturbations and inputs at the time of the perturbation. As a result, it will tend to specifically reproduce the perturbation itself, that is, the event that caused the complex changes in trajectory, and thus in reward.

## Gradient visualization

To better illustrate the effects of the rule, we run a simple experiment to compare the weight changes computed by our rule with those prescribed by several variants and other rules under the same conditions, isolating the effects of specific elements in the rules.

The experiment involves a recurrent network similar to those described in the previous sections. The task of the network is simply to determine whether a set of inputs have a positive or negative mean. In each episode, a vector of 10 randomly chosen inputs is presented for 100 ms, then (after a delay of 100 ms) the output neuron of the network must return 1 if the inputs have positive mean, or −1 if the inputs had negative mean, over the last 100 ms of the trial (thus each trial lasts 300 ms in total). The reward for each trial is computed as the mean negative absolute difference between neuron output and target response (−1 or 1) over the last 100 ms. At each episode, we apply a single perturbation (of random sign), at a fixed point in time within the response period. We then use

various plasticity rules, including the one proposed in this paper, to compute the appropriate change in weights (hereafter called the 'gradient'), and compare the gradients obtained from different rules. Note that no learning actually occurs: weights are randomly initialized at each trial, since we are only interested in comparing the gradients obtained under various rules.

*Figure 9a* plots the gradient obtained by the rule presented here, against the gradient obtained by the node-perturbation method, for many episodes (each with randomly generated weights). This reveals that the two rules produce largely similar gradients, with some warping introduced by the nonlinearity. Next we apply the rule described in this paper, but without supralinear amplification: the eligibility trace is now simply the accumulated produce of inputs by fluctuations, and this is then multiplied by the net reward. Note that this is equivalent to using the Exploratory-Hebbian method, but with a single reward value for each episode, rather than a continuous real-time reward signal. The computed gradients (*Figure 9b*) are now uncorrelated with the node-perturbation gradients, as expected from the reasoning presented above. This illustrates the crucial role of the supralinearity. To emphasize that this difference is caused by post-perturbation effects, *Figure 9c* uses the same method as *Figure 9b* (E-H with delayed rewards, no supralinear amplification), but now only accumulates the eligibility trace product over the 10 ms that follow the perturbation time. Since neurons have a time constant of 30 ms, this results in only partial relaxation. The gradients are now largely aligned with the node-perturbation gradients. If only 1 ms was used, the gradients would be identical to node-perturbation gradients and the points would fall exactly on the median line. As the time window is increased, relaxation effects eventually overwhelm the gradient and leave a residue dominated by noise, as in *Figure 9b*.

As a safety check, *Figure 9d* confirms that the full E-H method (with a real-time, continuous reward signal) correctly recovers the node-perturbation gradients in our experimental settings.

Finally, we emphasize that the choice of supralinear function is not crucial, as long as it is indeed supralinear. In *Figure 9e*, using a different supralinear function (the signed square function $f(x) = x.|x|$) produces largely similar result to *Figure 9a*. By contrast, in *Figure 9f*, using a sub-linear function (namely the square root) largely randomizes the obtained gradients, since the (large) perturbation effects are now suppressed with regard to (smaller) non-perturbation related effects.

## Discussion

This paper makes three contributions:

1- We introduce a biologically plausible learning algorithm that can train a recurrent neural network to learn flexible (context-dependent) tasks, using only time-sparse, delayed rewards and synapse-local information to guide learning.

2- We show that this rule can train networks for relatively complex tasks, requiring memory maintenance, selective attention, and coordination of multiple outputs.

3- We show that the trained networks exhibit features of neural activity observed in the primate higher cortex during similar tasks. In particular, we demonstrate highly dynamic population-wide encoding of task-relevant information, as observed in neural recordings (*Meyers et al., 2008*; *Stokes et al., 2013*; *Barak et al., 2010*); and we show that selective integration of sensory inputs occurs as described in both observational and modelling studies of primate prefrontal cortex during a similar selective attention task (*Mante et al., 2013*). In our view, the fact that these features of cortical activity arise spontaneously in networks trained with a biologically plausible rule (as opposed to training the network to directly reproduce observed neural activity traces) increases the plausibility of recurrent neural networks as a model of cortical computation, during both performance and learning of cognitive tasks.

Our proposed plasticity rule implements reward-modulated Hebbian learning between inputs, outputs, and rewards, with the crucial introduction of a supralinear amplification applied to the Hebbian plasticity increments (see Materials and methods). In other words, we posit that plasticity is dominated by large co-occurrences of inputs and outputs, while smaller ones are relatively ignored. This hypothesis of non-linear effects in Hebbian plasticity allows our rule to support robust learning in highly dynamic networks, without requiring non-Hebbian plasticity between segregated driving and perturbatory inputs (*Fiete et al., 2007*), or a continuous, real-time reward signal (*Legenstein et al., 2010*; *Hoerzer et al., 2014*) (see Materials and methods and Analysis). We note that this suggestion is similar to the independent proposal of so-called thresholded Hebbian rules

(**Soltoggio and Steil, 2013**), in which plasticity is only triggered if the Hebbian product reaches a certain threshold.

The flexible, dynamic coding observed in prefrontal activity has led to suggestions that cortex implements 'silent' memory traces by using short-term synaptic plasticity (**Barak et al., 2010**; **Stokes, 2015**). Short-term synaptic plasticity clearly plays an important role in neural responses, and may well play an important role in maintaining a 'hidden internal state' of the network (**Buonomano and Maass, 2009**). However, our network does not implement short-term synaptic plasticity; no weight modification occurs during the course of a trial (all learning occurs between trials), and all the decoding results reported above were obtained with frozen synaptic weights. Our results suggest that the highly dynamic activities spontaneously produced by near-chaotic recurrent networks can be harnessed to produce the dynamic encodings observed in experiments, using only sparse, delayed rewards and biologically plausible plasticity rules. Thus, while short-term synaptic plasticity clearly affects neural responses, it may not be required to explain the highly dynamic nature of working-memory encodings.

It is unlikely that cortical connectivity should be drastically and finely remodeled through a long training process for any new task. For example, while monkeys require extensive training to perform decision tasks, human subjects can quickly perform new tasks simply by verbal instruction. Rather, it is more likely that the process of slow, reward-modulated synaptic modification in cortical circuitry depicted here reflects the learning of functional networks capable of implementing a certain *type* of task (or cognitive ability), which must then be activated and parameterized for each instance of the task. The latter process of flexible task specification is likely to involve not just other cortical areas, but also the basal ganglia and dopamine system. Elucidating the interactions between cortical, limbic, and dopaminergic structures is an important future task for the study of flexible behavior and its neural implementation.

## Materials and methods

### Model description

Here we provide a full description of our model and proposed plasticity rule, with an emphasis on implementation details. In the Analysis section, we provide an extended discussion at a more intuitive level. Note that the source code for all simulations reported here is available online at http://github.com/ThomasMiconi/BiologicallyPlausibleLearningRNN.

### Network models

For most of our experiments, the model is a fully-connected continuous-time recurrent neural network of N neurons, governed by the classical RNN equations (**Sompolinsky et al., 1988**; **Sussillo and Abbott, 2009**; **Jaeger, 2001**; **Maass et al., 2002**):

$$\tau\frac{dx_i}{dt} = -x_i(t) + \sum_{j=1}^{N} J_{i,j}r_j(t) + \sum_{k=1}^{M} B_{i,k}u_k(t) \tag{3}$$

$$r_i(t) = tanh(x_i(t))$$

where $x_i$ is the excitation (or 'potential') of neuron i, $r_i$ is its response (or 'firing rate' / activity), $J_{i,j}$ is the connection weight from neuron j to neuron i, $u_k(t)$ is the current value of each of the M external inputs to the network, and $B_{i,k}$ is the connection weight from external input k to neuron i ($\tau$ is the relaxation time constant of neuron activation). J is initialized with weights taken from a normal distribution with mean 0 and variance $g^2/N$, while the input weights $B_{k,i}$ are fixed and taken from a uniform distribution over the $[-1,1]$ interval. Activations $x_i$ are initialized at the start of every trial with uniform noise in the $[-0.1, 0.1]$ range. For the simulations reported here, N = 200 (400 for the motor control task), $\tau$ = 30 ms, and g = 1.5. Note that the latter value places the networks in the early chaotic regime, where the long-term behavior generally remains non-periodic (**Sompolinsky et al., 1988**).

These canonical networks have signed responses and mix positive and negative weights, which is unrealistic. However, recently, Mastrogiuseppe and Ostojic have studied the conditions under which

chaotic ongoing activity can emerge in excitatory-inhibitory networks with nonnegative responses (*Mastrogiuseppe and Ostojic, 2016*). Using their results, we also implemented a network that produces ongoing chaotic activity with nonnegative responses, and separate populations of strictly excitatory and strictly inhibitory neurons (in accordance with Dale's law). As we demonstrate in the last section of Results, the rule can still learn cognitive tasks in this more realistic network. In these simulations, there are 100 excitatory and 100 inhibitory neurons. The neuron response function is a nonnegative piecewise-linear function:

$$
\begin{aligned}
r_i(t) &= 0 && \text{if } x_i(t) < -m \\
r_i(t) &= x_i(t) + m && \text{if } x_i(t) > -m \text{ and } x_i(t) - m \\
r_i(t) &= M && \text{if } x_i(t) - m
\end{aligned}
$$

We set $-m = -2$ and $M = 20$. Note that the network is largely non-saturating: neural responses usually do not reach responses close to the maximum $M$, which is another similarity with cortical networks. The connection matrix is initialized as semi-sparse: each neuron receives connections from 50 randomly chosen excitatory neurons and 50 randomly chosen inhibitory neurons. All excitatory connections have initial weight 1. All inhibitory connections have initial weight $-g_{inhib} = -1.2$. As learning occurs, connection weights change from their initial values; however, connections from excitatory neurons are always clipped to 0 from below, while connections from inhibitory neurons are always clipped to 0 from above. The network otherwise operates as described above (in particular, the differential equation governing $x_i(t)$ is unchanged).

In all simulations, four arbitrarily chosen neurons have a constant activation x = 1 and thus provide a bias input to other neurons. There is no separate feedback or output network. Instead, one or more neurons in the network are arbitrarily designated as the 'output' neurons, and their responses at any given time are used as the network's response (these neurons are otherwise identical to all others).

## Learning rule

Synapses between neurons are modified according to a novel form of reward-modulated Hebbian learning, which we now describe.

First, in order to produce exploratory variation in network responses across trials, each neuron in the network occasionally receives a random perturbation $\Delta_i(t)$ to its activation; these perturbations are not segregated from 'normal' inputs (in contrast to *Fiete et al., 2006*, *2007*). Note that $\Delta_i(t)$ might also represent random noise, or a 'teaching' signal from a different area. Also, perturbations are applied in all simulations reported here, both at learning and testing/decoding time (perturbations correspond to the sharp spikes in the curves in *Figure 1c*, which would otherwise show only smooth curves; they also cause the spread of dots around their central values in *Figure 3*). In this paper, $\Delta_i(t)$ is taken from a uniform distribution within the [−0.5, 0.5] range, occurring randomly and independently for each neuron with a mean rate of 3 Hz (rates of 1 Hz or 10 Hz also give satisfactory results).

During a trial, at every time step, every synapse from neuron *j* to neuron *i* accumulates a *potential* Hebbian weight change (also called eligibility trace [*Izhikevich, 2007*]) according to the following equation:

$$ e_{i,j}(t) = e_{i,j}(t-1) + S\big(r_j(t-1) * (x_i(t) - \bar{x}_i)\big) $$

Remember that $r_j$ represents the output of neuron *j*, and thus the current input at this synapse. $x_i$ represents the activation of neuron *i* and $\bar{x}_i$ represents a short-term running average of $x_i$, and thus $x(t) - \bar{x}$ tracks the fast fluctuations of the neuron's output. Thus this rule is essentially Hebbian, based on the product of inputs and output (fluctuations). Crucially, *S* is a monotonic, supralinear function of its inputs; in other words, we posit that the plasticity mechanism is dominated by large increments, and tends to suppress smaller ones. The particular choice of *S* is not critical, as long as it is supralinear. In this paper we simply used the cubic function $S(x) = x^3$. Sign-preserving squaring $S(x) = x|x|$ also gives satisfactory results; however, simply using the identity function fails to produce learning. The supralinear amplification of co-occurrences allows our learning rule to successfully learn from instantaneous deviations of activity, using only sparse, delayed rewards, without requiring a continuous, real-time reward signal (*Legenstein et al., 2010*; *Hoerzer et al., 2014*); see Discussion and Analysis.

Note that the eligibility trace for any synapse is accumulated over the course of a trial, with each new timestep adding a small increment to the synapse's eligibility trace / potential weight change.

At the end of each trial, a certain reward R is issued to the network, based on the network's performance for this trial as determined by the specific task. From this reward, the system computes a *reward prediction error* signal, as observed in physiological experiments, by subtracting the expected reward for this trial $\bar{R}$ (see below for computation of $\bar{R}$ $\bar{R}$) from the actually received reward R. This reward-prediction signal is used to modulate the eligibility trace into an actual weight change:

$$\Delta J_{i,j} = \eta e_{i,j}(R - \bar{R})$$

where η is a learning rate constant, set to 0.5 for all simulations described here.

To compute the reward prediction error signal (R-$\bar{R}$), we need to estimate the expected reward in the absence of perturbation, $\bar{R}$. Following (*Frémaux et al., 2010*), we simply maintain a running average of recent rewards for trials of the same type (where trial type is determined by the combination of inputs). As (*Frémaux et al., 2010*) pointed out, it is important that separate traces should be maintained for each trial type, so as to provide an accurate estimation of the expected reward $\bar{R}$ for each trial. Thus, after the *n*-th trial of a given type, $\bar{R}$ is updated as follows:

$$\bar{R}(n) = \alpha_{trace}\bar{R}(n-1) + (1 - \alpha_{trace})R(n)$$

Where R(n) is the reward for this trial, and $\bar{R}$(n-1) was the expected reward after the previous trial of the same type. In all simulations, $\alpha_{trace}$ = 0.33.

To stabilize learning, we clip the weight modifications for each trial to have a maximum absolute value of $10^{-4}$ (across experiments, roughly 10% of all potential weight modifications exceed this value and are clipped).

## Description of tasks

### Delayed nonmatch-to-sample task

The first task considered here is a simple delayed nonmatch-to-sample problem (*Figure 1*). In every trial, we present two brief successive inputs to the network, with an intervening delay. Each input can take either of two values, labelled A and B respectively. The task is to determine whether the two successive inputs are identical (AA or BB), in which case the network should output −1; or different (AB or BA), in which case the network should output 1. We specify the input stimuli by using two different input channels u1 and u2; the identity of the input stimulus is determined by which channel is activated (i.e., for stimulus A, u1 = 1 and u2 = 0; for stimulus B, u1 = 0 and u2 = 1; remember that each input channel $u_k$ is transmitted to the network by its own independent set of weights - see Model Description above). In every trial, the first stimulus is presented for 200 ms, then after a 200 ms delay the second stimulus is presented for 200 ms. Outside of input presentation periods, both input channels are set to 0. The trial goes on for an additional 400 ms, thus each trial is 1000 ms long. The network's overall response is determined by the activity of the arbitrarily chosen output neuron over the last 200 ms of the trial (the so-called 'response' period). The overall error for this trial is the average *absolute* difference between the network's output (that is, the activity of the output neuron) and the target response (1 or −1 depending on presented stimuli), over these last 200 ms.

For details on how the network activity was analyzed for *Figures 2* and *3*, see below.

### Selective integration of context-cued sensory inputs

This task was introduced by Mante and colleagues (*Mante et al., 2013*). In this study, monkeys looked at randomly-moving colored dots, in which both the value and coherence of motion direction and dot color varied from trial to trial. Monkeys had to report the dominant motion direction, or the dominant color, according to current task conditions; thus, the same stimulus could entail different appropriate responses depending on current context. Furthermore, due to the noisy stimulus, the task required temporal integration of sensory input. Importantly, the authors showed that prefrontal neurons registered inputs from both the relevant and the irrelevant modality; however, inputs from the irrelevant modality had no long-term impact on neural activities, while inputs from the relevant

modality were selectively integrated over time. Thus, only information from the relevant modality contributed to the final decision.

Our settings are deliberately similar to those described by Mante, Sussillo and colleagues in their neural network implementation of the task, and Song and colleagues in their own implementation (*Song et al., 2016*). The network has two 'sensory' inputs (representing the two stimulus modalities of motion and color) and two 'context' inputs (which specify which of the two modalities must be attended to). The sensory inputs are noisy time-series, centered on a specific mean which indicates the 'value' of this input for this trial. More precisely, each of the two sensory inputs is sampled at each time step from a Gaussian variable with variance 1 and a mean, or bias, randomly set at either −0.5 or 0.5 for each trial (this is for the learning phase; for the testing phase used to generate the psychometric curves in *Figure 4*, the bias is varied in increments of 0.1 from −0.5 to 0.5, inclusive). The mean/bias of the Gaussian (positive or negative) represents the 'direction' or 'value' of the corresponding sensory input (left vs. right, or red vs. green). The context inputs are set to 1 and 0, or 0 and 1, to indicate the relevant modality for this trial. The goal of the network is to determine whether the sensory input in the relevant modality has positive or negative mean.

Sensory inputs are presented for the first 500 ms of the trial, followed by a 200 ms response period during which all sensory inputs are set to 0. The expected response of the network is 1 if the relevant sensory input has a positive mean, and −1 otherwise; thus the same sensory input can entail different appropriate responses depending on the context. As for the previous task, the network's response for a trial is the firing rate of the arbitrarily chosen output cell, and the error for a trial is the average absolute difference between the firing rate of this output cell and the appropriate response for this trial (either −1 or 1) over the 200 ms response period.

For details of network analysis (*Figure 5*), see below.

## Analysis of network activity

### 1- Decoding of network information in a delayed nonmatch-to-sample task

In the delayed nonmatch-to-sample task, we used a cross-temporal classification analysis (*Meyers et al., 2008*; *Stokes et al., 2013*) to investigate how fully trained networks encode information over time (*Figure 2*). The interpretation of these cross-temporal decoding accuracy matrices is that they tell us not only whether the network encodes a certain task-relevant variable, but also whether it uses similar representations to encode this variable at different points in time. If we train one such classifier using data at time $t$ in some trials, and then use it to decode population activity from data at the same time $t$ in other trials, then decoding accuracy measures how strongly the network encodes the feature at that time point $t$. However, when the decoder is trained on data at time $t_{learn}$ and then applied to population activity data at time $t_{decode}$, the resulting accuracy measures the stability in the network's 'neural code' for this feature across both time points, i.e., how similarly the decoded feature is represented by the network across these time points. If representations are similar across both time points (that is, if the network use similar patterns to represent each possible value of the feature across both time points), then classifiers successfully trained with population activities at time $t_{learn}$ should also produce accurate decoding of population activities at time $t_{decode}$. By contrast, if the network uses different representations/encoding of task features at these two time points, cross-temporal accuracy should be poor; this should be represented as 'bottlenecks' of high accuracy on the cross-temporal decoding plots, whereby information is high along the diagonal (i.e. the feature is indeed encoded by the network at that given time), but away-from-diagonal (cross-temporal) decoding accuracy is low. This is precisely what we observe in *Figure 2*.

We follow the maximal-correlation classifier approach described in *Meyers et al. (2008)* as closely as possible. Briefly, we want to measure how well a certain task-relevant feature (identity of first presented stimulus, or identity of second presented stimulus, or final response) can be predicted by observing network activity at time $t_1$, using a classifier trained on network activity at time $t_2$. First, we sample the activation of each neuron, every 10 ms, for each trial. This data is stored in a matrix of 100 rows and 200 columns, indicating the activities (firing rates) of all 200 neurons at each of the 100 sampling times. We first generate 80 trials (20 per possible condition, where 'condition' is defined as one of the four possible stimulus combination: AA, AB, BA or BB) with a trained network. The time course of neural activity will differ somewhat between successive trials, even for identical conditions, due to noise. Then we iterate the following procedure. For each of all four possible

conditions, we randomly choose half the trials as 'training' trials, and the other half as 'testing' or 'decoding' trials. The training trials corresponding to the same category that we are trying to decode (for example, all stimuli having the same first presented stimulus) are averaged together, pointwise, for each neuron and each time point, giving a 'prototype' matrix of activation for each neuron at each timepoint under this category. This training data allows us to decode the category of each testing trial, at each point in time, using maximum-correlation classification, in the following way. We compute the Pearson correlation between each row of each 'testing' trial and each row of each 'prototype' trial. Each such correlation between row $i$ of a testing trial and row $j$ of a training category-average tells us how much the population activity at time $i$ in the testing trial resembles the average population activity at time $j$ for this particular category. We can then select the category for which this correlation is maximal, at each training/testing timepoint pair, as the 'decoded' category for each testing trial. For each testing trial, this provides a 100 × 100 matrix of decoded categories (one for each pair of training and testing timepoints). Of course, each testing trial belongs to only one category, so only one possible answer is correct, and thus we can compute another 100 × 100 matrix of binary values, indicating whether the decoded category at a given point in the decoding matrix (i.e., for any given pair of training and testing timepoints) is correct. The average of these 'correctness matrices', over all testing trials, provides the accuracy in cross-temporal decoding of this category for every training/testing pair of timepoints. We iterate this whole procedure 100 times and average together the resulting 'correctness' matrices. The resulting 100 × 100 matrix indicates at each row $i$ and column $j$ the proportion of times that the decoded category for population activity at timepoint $j$ was correct, using training data from timepoint $i$. This is the matrix shown in each of the panels in *Figure 2* (one for each of the three categories to be decoded).

## 2- Orthogonal decoding of network information during a selective integration task

For the selective integration task, we used the analysis method introduced by Mante, Sussillo and colleagues (*Mante et al., 2013*), and also used by Song and colleagues (*Song et al., 2016*) (see *Figure 5*). Intuitively, the purpose of this method is to estimate how much information the network encodes about different task feature (input value, context, final choice, etc.) *independently* from each other.

After generating multiple trials under various conditions (context – that is, relevant modality – and bias for each modality) with a fully trained network, we regress the activity of each neuron over the values of features of interest (context, value of each modality, and final choice) for each trial. This gives us a set of weights for each neuron, one for each feature, representing how much each feature influences the neuron's firing rate. We then 'switch views' by grouping together all such weights for any given feature (200 weights - one per neuron). This in turn produces vectors in neuron population space, along which the feature is in a sense maximally represented (notice that this is quite different from, and not equivalent to, the simpler idea of simply regressing each feature over the firing rates of the neurons across trials). We then orthogonalize these vectors using QR decomposition, to ensure that these representations are as independent from each other as possible. Projecting population activity at a given time over the resulting vectors approximates the network's current estimate of the corresponding feature at that time. For successive time slices, we average network activity vectors corresponding to the same value of bias in a certain modality, a certain attended modality, and a certain final choice. We refer the reader to *Mante et al. (2013)* for a complete description of the method.

We project population activity, averaged within various groups of trials, at each point in time, over these decoding axes. The trials are grouped according to final choice, value of one modality (either modality 1 or modality 2), and current context (i.e., relevant modality), and the population activity at each point in time is averaged across all trials within each group. When the resulting averages are projected over the orthogonal feature vectors, they produce trajectories, indicating the network's encoded value for each feature, at each point in time, for trials of this group. Only correct trials are used, and thus certain combinations are impossible (for example, positive value of modality 1 bias, while attending modality 1, with a final choice of −1); this is reflected in the top-left and bottom-right panels of *Figure 6*, which contain half as many trajectories as the top-right and bottom-left panels.

## Acknowledgements

We thank W Einar Gall for useful comments and suggestions. We thank Vishwa Goudar for helpful discussions. We thank H Francis Song for important insight regarding the computation of state-space trajectories in *Figure 5*. This work was supported by the Neurosciences Research Foundation through funding from The G Harold and Leila YMathers Charitable Foundation and the William and Jane Walsh Charitable Remainder Unitrust, for which we are grateful.

## Additional information

### Funding

| Funder | Author |
| --- | --- |
| G Harold and Leila Y. Mathers Foundation | Thomas Miconi |
| The William and Jane Walsh Charitable Remainder Unitrust | Thomas Miconi |

The funders had no role in study design, data collection and interpretation, or the decision to submit the work for publication.

### Author contributions

TM, Conceptualization, Software, Formal analysis, Investigation, Methodology, Writing—original draft, Writing—review and editing

### Author ORCIDs

Thomas Miconi, http://orcid.org/0000-0002-7897-4492

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
