## [Decision Letter]

Thank you for submitting your article "Biologically plausible learning in recurrent neural networks reproduces neural dynamics observed during cognitive tasks" for consideration by *eLife*. Your article has been reviewed by two peer reviewers, and the evaluation has been overseen by Michael Frank as Reviewing Editor and Timothy Behrens as the Senior Editor. The reviewers have opted to remain anonymous.

The reviewers have discussed the reviews with one another and the Reviewing Editor has drafted this decision to help you prepare a revised submission.

Summary:

The manuscript is a theoretical/computational study of reward-based learning in recurrent networks of (non-spiking) neurons. The author compares the simulation results to previously published analyses of experimental data. The author describes an improved reward-based learning rule which is more biologically plausible than other supervised learning rules that have previously been used for similar purposes. The author applies the rule to a number of relevant tasks. It is shown that a network equipped with the rule can learn these tasks surprisingly well. Further, it is demonstrated that the resulting network dynamics is consistent with experimental data.

Essential revisions:

1) Both Reviewers were enthusiastic about the contribution, which they agreed was the learning rule itself, rather than its ability to produce dynamics, but both expressed concerns that it wasn't sufficiently clear why the rule works. This point was underscored in the consultation session in which it was noted that without further analysis, there is a strong risk that either i) the learning rule will be found to break down for more complex tasks and generality will be lost, or ii) someone else will soon publish a detailed theoretical study of this learning rule and greatly shadow this contribution. Thus the most important point for revision is to focus efforts on quantifying the intuitions given in the supplementary material.

Here are more specific comments and suggestions in this regard:

On the one hand, it is very nice to see a simple, plausible learning rule (in a fairly well-written paper) learn non-trivial computations in *recurrent* networks where we know the gradients of the objective function are hopelessly non-local. I see this paper as a much welcome addition to the upcoming literature on reverse-engineering brain computations by training RNNs. On the other hand, the author seems to spend comparatively too much effort on "extra goodies", or what I would see as mainly "asides" (e.g. studying the activity of those networks after training), and not enough in trying to understand the reasons why this learning rule works so well (again, I see the learning rule itself as the main contribution of this paper - will other reviewers agree?). Dissecting the inner workings of the proposed learning rule quantitatively would considerably strengthen the paper, and would also suggest situations where one would expect it to fail (in a way, the paper could have been more "self-critical"). In its current formulation, the learning rule looks like a hack - so, as for any hack, one has little reason to believe it is generally applicable. One could still argue that the range of tasks studied here is wide enough to suggest this rule will robustly apply to other scenarios, but I would be more convinced by some more thorough theoretical analysis. The supplementary material does give some reasonable intuitions (especially regarding the nonlinearity in the rule), but they remain hand-wavy and at the very least should be included in the main text, since it is central to the paper.

Here are a couple of suggestions:

i) Substantiate the relatively vague intuitions given in the supplementary material, by numerically and/or analytically quantifying the most important relationships stated there ("x is greater than y, while z is neglible etc.").

ii) To dissect the role of the supralinear amplification of co-fluctuations, I would have started by much simpler RNN problems, such as simple nonlinear regression in a multi-layer, *feedforward* network - this would get rid of all the subtleties related to recurrent processing, short-term memory etc., which I found the supplementary material does not handle very well. How do the updates prescribed by the proposed learning rule compare with the exact gradient of the reward function? How does that depend on supralinear amplification?

iii) Somewhat more generally - we know some of the reasons why it is hard to train recurrent neural networks (without extra hacky components like LSTM etc.): vanishing/exploding gradients, very ill-conditioned saddle points, etc. How does the proposed learning rule overcome these difficulties? (and in fact, does it? - are the tasks studied here representative of more complex cognitive tasks with respect to these generic difficulties in training RNNs?).

2) Figure quality could be improved. Sub panels with labels (A, B, etc.) should be established and referred to. Figure 3 is ugly.

3) The description of Figure 5 in the legend and in the text is hard to follow. For example, in Figure 5, the meaning of colors should be explained.

4) Compared to experiments, the delay of 200 ms in task 1 is very short. Does it also work with more realistic delays on the order of seconds? Relatedly, where the learning rule might break down, it may be nice to address explicitly in discussion whether this rule is dependent on a fixed interval between task relevant events. One central problem for RL rules that depend on eligibility traces with fixed time constants is that without other mechanisms, these may have difficulty in tasks that have distractors, variable timing, or number of intervening events between task relevant stimuli (see for example O'Reilly & Frank 2006 Neural Computation).

5) Although it is common to use rate-based networks in such models, the model used contains a number of coarse approximations to biological reality.i) Rate-based neurons are used instead of spiking models.ii) The tanh-activation function allows for negative neuron outputs. What happens for non-negative outputs (e.g. a logsig)?iii) Synaptic weights from one presynaptic neuron can be positive or negative (violating Dale's law) and presumably even change the sign during learning. What if more realistic constraints are included?

It would be nice if one could have at least one control simulation where ii) and iii) are considered and a brief discussion. This would strengthen the main claims.

6) It would be interesting to discuss how the learning rule compares to the supervised approach considered in other works? How much does one lose in terms of performance or convergence speed?

7) The eligibility trace is typically decaying. According to Equation 3 it is not in this model. Why, and is it important?

[Editors' note: further revisions were requested prior to acceptance, as described below.]

Thank you for resubmitting your work entitled "Biologically plausible learning in recurrent neural networks for cognitive tasks" for further consideration at *eLife*. Your revised article has been favorably evaluated by Timothy Behrens as Senior editor), Michael Frank as Reviewing editor, and two reviewers.

The manuscript has been improved but there are some remaining issues that need to be addressed before acceptance, as outlined below:

The revised paper is clearer in how it exposes the learning rule, but reviewers differed as to whether this was satisfactory or not. Please see concerns below, if you can respond directly to them and expand the manuscript to address them the paper will be improved.

Reviewer #1:

I think this revision only partially addresses my original concern. The supplementary material has merely been rebranded "appendix", with a few (nice) additions, where I was expecting a convincing theoretical analysis of the learning rule as a first-class citizen in the main text. Why does this learning rule work? Why does learning success rely on moments of higher order than the usual Hebbian covariances? I am not sure I understand it better now after reading this revision.

The comparison to the exact gradient in Figure 9 is a nice starting point, but there is potential to squeeze more out of this toy example to explore the role of the supralinear gain function more systematically; e.g. how does the alignment between proposed rule updates and true gradient depend on the exact type of superlinear function used? (e.g. on the exponent n>1 if x → x is used).

The addition of Figure 8 and the explanation of the relaxation effect is not entirely convincing to me. In paragraph two of subsection “Removing the need for continuous reward signals with supralinear amplification”, all the author is saying here is that the average of xx¯ is zero - sure - but this is not what the eligibility trace is accumulating: it's estimating a covariance. The argument works OK when the input is constant, but in recurrent nets it will be anything but constant. In this respect, I am not sure that the example of Figure 9 (which uses constant inputs) is representative. Can the author comment?

I still find the argument appealing though, if it can be made more precise/quantitative; in particular, does this argument inform us of what "a good time constant" would be for the running average of x - e.g. that it should be larger than the typical autocorrelation time of the input fluctuations? Could another fix be to remove a running average of *r_j_* from the input *r_j_* too, so that the pre and post "relaxation effects" cancel?

Reviewer #2:

The manuscript has significantly been improved in this revision. In particular the learning rule has been analyzed more thoroughly, which provides important insights to the general learning problem addressed in this paper. The author has also added new experiments that show that the rule also works under some more challenging setups. The discussion of the learning rule is now in the Appendix, which is not optimal but acceptable. I think no additional work is necessary at this point.

---

## [Author Response]

*Essential revisions:*

*1) Both Reviewers were enthusiastic about the contribution, which they agreed was the learning rule itself, rather than its ability to produce dynamics, but both expressed concerns that it wasn't sufficiently clear why the rule works. This point was underscored in the consultation session in which it was noted that without further analysis, there is a strong risk that either i) the learning rule will be found to break down for more complex tasks and generality will be lost, or ii) someone else will soon publish a detailed theoretical study of this learning rule and greatly shadow this contribution. Thus the most important point for revision is to focus efforts on quantifying the intuitions given in the supplementary material.*

*Here are more specific comments and suggestions in this regard:*

*On the one hand, it is very nice to see a simple, plausible learning rule (in a fairly well-written paper) learn non-trivial computations in recurrent networks where we know the gradients of the objective function are hopelessly non-local. I see this paper as a much welcome addition to the upcoming literature on reverse-engineering brain computations by training RNNs. On the other hand, the author seems to spend comparatively too much effort on "extra goodies", or what I would see as mainly "asides" (e.g. studying the activity of those networks after training), and not enough in trying to understand the reasons why this learning rule works so well (again, I see the learning rule itself as the main contribution of this paper - will other reviewers agree?).*

We agree that the rule requires better description and explanation, and we have implemented the reviewers’ recommendations in that regard, both by improving the discussion and adding quantitative experiments (see below). On the other hand, clearly different readers will be interested in different aspects. In our opinion, the fact that the networks can not only solve those (admittedly simple) tasks, but also reproduce some of the puzzling dynamics of real cortical networks, is interesting in itself and warrants discussion. We have certainly seen some interest in these results from informal communications and presentations in poster form at conferences.

We did change the title. The previous title was “Biologically plausible learning in recurrent neural networks reproduces neural dynamics observed during cognitive tasks”. The new title is “Biologically plausible learning in recurrent neural networks for cognitive tasks”, which concentrates on the important point (as pointed out by the reviewer) and is less potentially contentious.

*Dissecting the inner workings of the proposed learning rule quantitatively would considerably strengthen the paper, and would also suggest situations where one would expect it to fail (in a way, the paper could have been more "self-critical"). In its current formulation, the learning rule looks like a hack - so, as for any hack, one has little reason to believe it is generally applicable. One could still argue that the range of tasks studied here is wide enough to suggest this rule will robustly apply to other scenarios, but I would be more convinced by some more thorough theoretical analysis. The supplementary material does give some reasonable intuitions (especially regarding the nonlinearity in the rule), but they remain hand-wavy and at the very least should be included in the main text, since it is central to the paper.*

We agree with the reviewer. Following the reviewer’s request, we have moved the detailed discussion of the rule into the main text (Appendix), streamlined it, and added quantitative experiments described below.

*Here are a couple of suggestions:*

*i) Substantiate the relatively vague intuitions given in the supplementary material, by numerically and/or analytically quantifying the most important relationships stated there ("x is greater than y, while z is neglible etc.").*

*ii) To dissect the role of the supralinear amplification of co-fluctuations, I would have started by much simpler RNN problems, such as simple nonlinear regression in a multi-layer, feedforward network - this would get rid of all the subtleties related to recurrent processing, short-term memory etc., which I found the supplementary material does not handle very well. How do the updates prescribed by the proposed learning rule compare with the exact gradient of the reward function? How does that depend on supralinear amplification?*

We agree with the reviewer’s observations in both of these points. We have tried to implement the reviewer’s recommendations by implementing a simple experiment (single neuron, time-constant inputs, single perturbation, iterated over many randomly chosen weights and inputs). We use this simple experiment to compare the gradient obtained by the proposed rule with gradients obtained under various other methods, including the gradient obtained by supervised backpropagation as a “ground truth” (Figure 9). This experiment seems to confirm that the supralinear amplification is indeed the crucial element that allows the proposed rule to recover the correct direction of gradients (compare Figure 9). We also see additional evidence to support the idea that this is caused by relaxation effects (Figure 9).

*iii) Somewhat more generally - we know some of the reasons why it is hard to train recurrent neural networks (without extra hacky components like LSTM etc.): vanishing/exploding gradients, very ill-conditioned saddle points, etc. How does the proposed learning rule overcome these difficulties? (and in fact, does it? - are the tasks studied here representative of more complex cognitive tasks with respect to these generic difficulties in training RNNs?).*

The tasks here are relatively simple in comparison to the much more complex tasks that supervised RNNs are applied to these days (natural language processing, etc.) At any rate, as explained in the paper, the rule is an implementation of a specific algorithm that is commonly used in machine learning, namely, the REINFORCE algorithm, so we would expect performance to be on par with it – accounting for the specific characteristics of the networks discussed here (persistent chaotic activity, nontrivial time constants, etc.).

*2) Figure quality could be improved. Sub panels with labels (A, B, etc.) should be established and referred to. Figure 3 is ugly.*

We have added labels to all panels and tried to de-uglify Figure 3 (and others!)

*3) The description of Figure 5 in the legend and in the text is hard to follow. For example, in Figure 5, the meaning of colors should be explained.*

Figure 5 describes a complex, multi-step decoding procedure from Mante, Sussillo et al. Nature 2013. We have tried to clarify both the caption and the description in the main text, though the procedure is quite involved. We did add a description for the meaning of colors.

*4) Compared to experiments, the delay of 200 ms in task 1 is very short. Does it also work with more realistic delays on the order of seconds? Relatedly, where the learning rule might break down, it may be nice to address explicitly in discussion whether this rule is dependent on a fixed interval between task relevant events. One central problem for RL rules that depend on eligibility traces with fixed time constants is that without other mechanisms, these may have difficulty in tasks that have distractors, variable timing, or number of intervening events between task relevant stimuli (see for example O'Reilly & Frank 2006 Neural Computation).*

We agree with the reviewer that this is an important subject. To test this, we performed additional experiments with the delayed non-match to sample task. Results are shown in the new Figure 7. We first increased the delay to 1000 ms (Figure 7). Then we tried variable delays (from 300 to 800ms Figure 7).

Basically, the rule is fine with longer delays. It does take a hit with variable delays, requiring significant reduction in learning rate to deal with the increased variance; however, it still successfully learns the task – eventually.

*5) Although it is common to use rate-based networks in such models, the model used contains a number of coarse approximations to biological reality.i) Rate-based neurons are used instead of spiking models.ii) The tanh-activation function allows for negative neuron outputs. What happens for non-negative outputs (e.g. a logsig)?iii) Synaptic weights from one presynaptic neuron can be positive or negative (violating Dale's law) and presumably even change the sign during learning. What if more realistic constraints are included?*

*It would be nice if one could have at least one control simulation where ii) and iii) are considered and a brief discussion. This would strengthen the main claims.*

We have actually added an experiment with a network using non-negative responses and separate E and I neurons (i.e. observing Dale’s law), as shown in Figure 7. The rule still manages to learn the task with these more realistic networks.

Note that this would not have been possible without the recent work of Mastrogiuseppe and Ostojic (cited in the paper). The whole paper relies on the strong theoretical guarantees developed for the canonical RNN model by Sompolinsky et al. 1993 – in particular, a well-defined regime in which ongoing non-periodic (chaotic) activity is guaranteed.

These guarantees are less important when using supervised learning, which will fit the weights to reproduce any trajectory (as in HF Song et al. Plos Comp Biol 2016). But they are crucial when using reinforcement learning which assumes that the network can already generate the proper”kind” of trajectories in the first place. Mastrogiuseppe and Ostojic showed that a similar chaotic regime can be found and parametrized for networks with nonnegative responses and separate E and I neurons.

Furthermore, they also show that the network in this regime are non-saturating (responses don’t reach the maximum value), which may be of interest to the reviewer (see below).

*6) It would be interesting to discuss how the learning rule compares to the supervised approach considered in other works? How much does one lose in terms of performance or convergence speed?*

Supervised learning is quite a different problem – it assumes that we already have a good trajectory and want to find the weights that reproduce this trajectory. The problem considered here is the more general problem of reinforcement learning, where we don’t know what the good trajectory is

we only have a trajectory evaluator that tells us whether a given trajectory is “better” or “worse”, without telling us what part of the trajectory makes it so. Supervised learning is of course faster and more reliable than reinforcement learning if you can apply it – that is, if you already know the correct trajectory.

*7) The eligibility trace is typically decaying. According to Equation 3 it is not in this model. Why, and is it important?*

As discussed in the Introduction and Appendix, the rule proposed here inherits much from Fiete and Seung 2006, Legenstein et al. 2010 and Hoerzer et al. 2014. All of them use a non-decaying eligibility trace. Izhikevich 2007 used a time constant of 1 sec, which should have very little effect in the present paper where all trials last 1s or less.

Just to be sure, we tried the DNMS task with a decaying eligibility trace with a 1-second time constant, and as expected it worked very similarly.

[Editors' note: further revisions were requested prior to acceptance, as described below.]

*The revised paper is clearer in how it exposes the learning rule, but reviewers differed as to whether this was satisfactory or not. Please see concerns below, if you can respond directly to them and expand the manuscript to address them the paper will be improved.*

*Reviewer #1:*

*I think this revision only partially addresses my original concern. The supplementary material has merely been rebranded "appendix", with a few (nice) additions, where I was expecting a convincing theoretical analysis of the learning rule as a first-class citizen in the main text. Why does this learning rule work? Why does learning success rely on moments of higher order than the usual Hebbian covariances? I am not sure I understand it better now after reading this revision.*

We have moved the analysis of the learning rule into an “Analysis” section just after the Results section. We have also tried to clarify and expand the main points of this analysis, and updated the experiments in this section with a more complex, representative setting (see below).

*The comparison to the exact gradient in Figure 9 is a nice starting point, but there is potential to squeeze more out of this toy example to explore the role of the supralinear gain function more systematically; e.g. how does the alignment between proposed rule updates and true gradient depend on the exact type of superlinear function used? (e.g. on the exponent n>1 if x → x^n^ is used).*

We agree with the reviewer. We have added two more examples: one with a different supralinear function, namely “sign-preserving square” (x*|x|) which was already mentioned in the methods as satisfactory, and one with a sublinear function (square root). As expected, the former shows very similar results to the main rule, while the latter fails to recover correct gradients.

*The addition of Figure 8 and the explanation of the relaxation effect is not entirely convincing to me. In paragraph two of subsection “Removing the need for continuous reward signals with supralinear amplification”: all the author is saying here is that the average of*
xx¯
*is zero - sure - but this is not what the eligibility trace is accumulating: it's estimating a covariance. The argument works OK when the input is constant, but in recurrent nets it will be anything but constant.*

In this very simple example, the total eligibility trace, being the accumulated product of inputs by output fluctuations, will be zero because the inputs are constant, as the reviewer notes below.

If the inputs are not constant, the eligibility trace will include both the product of the perturbation effect by the inputs at the time of the perturbation (which is what we want), AND the product of immediately subsequent inputs by the relaxation terms (which is absolutely not what we want). The latter part adds “garbage” components to the eligibility trace. Furthermore, if there is any temporal correlation in successive inputs, then these “garbage” components will tend to partially cancel out the perturbation-related terms (because they will be the product of similar inputs by an output relaxation term that has the opposite sign as the output perturbation term). This, we suggest, causes the inability of the E-H rule to learn without continuous, real-time reward signal (experiments in the new Figure 9 are meant to confirm this interpretation).

We have expanded and clarified our explanation of this point, which is crucial for our discussion.

We are unsure whether the eligibility trace can be said to “compute a covariance”, either formally or intuitively. Note that the inputs are *not* centered or detrended in any way, and neither should they be.

Intuitively, we think of the eligibility trace as a gradient: the quantity that should be added to the weights in order to replicate the encountered perturbations (or at least, move outputs in the same direction as the perturbations) when the same inputs are presented again in the future. It serves this role, formally, in the original REINFORCE algorithm (See Section 4, especially Equation 9 in William 1992).

*In this respect, I am not sure that the example of Figure 9 (which uses constant inputs) is representative. Can the author comment?*

We agree with the reviewer that the simple feedforward network with constant inputs shown in the previous version of Figure 9 may not be a sufficient test of the various methods. Therefore, we have replaced this with a more complex experiment using a fully recurrent network, similar to the one used in the other experiments of the paper. In these settings, the inputs (which are the activities of the neurons in the network) are constantly varying. We show that the computed gradients are very similar and produce the expected results.

Note that we have also added a simulation of the full Exploratory-Hebbian rule, confirming that it does recover the correct gradients when allowed to use of a real-time, continuous reward signal (Figure 9).

I still find the argument appealing though, if it can be made more precise/quantitative; in particular, does this argument inform us of what "a good time constant" would be for the running average of x - e.g. that it should be larger than the typical autocorrelation time of the input fluctuations?

A long running average on the activation would start extracting slow,

non-perturbation-related fluctuations to a significant degree, which would defeat the point. This is in addition to the necessary temporal smearing. Our early experiments suggested that shorter time constants worked best.

*Could another fix be to remove a running average of r_j_ from the input r_j_ too, so that the pre and post "relaxation effects" cancel?*

Again, we stress that in our rule (as in node-perturbation or REINFORCE), we must include the full inputs *r_j_*, not their fluctuations. The inputs are used “as is”, which is exactly how it should be. If we used input fluctuations rather than raw inputs, we would add the wrong quantity to the weights.

To take a very simple example, imagine that a positive perturbation is applied while the input at the synapse is high and positive, but declining. If we want to reproduce the effect of the perturbation next time this input is shown, we should add a positive quantity to the weight. This is exactly what node-perturbation, REINFORCE, and our rule suggest – but using input fluctuations rather than raw inputs in the eligibility trace would actually force us to add a negative quantity to the weights, which would actually make subsequent responses lower, i.e. opposite to the direction of the perturbation.